# Temporal Score Rescaling
# for Temperature Sampling in Diffusion and Flow Models

**Yanbo Xu** [* 1]  **Yu Wu** [* 1]  **Sungjae Park** [1]  **Zhizhuo Zhou** [2]  **Shubham Tulsiani** [1]

https://temporalscorerescaling.github.io

## Abstract

We present a mechanism to steer the sampling diversity of denoising diffusion and flow matching models, allowing users to sample from a sharper or broader distribution than the training distribution. We build on the observation that these models leverage (learned) score functions of noisy data distributions for sampling and show that rescaling these allows one to effectively control a 'local' sampling temperature. Notably, this approach does not require any finetuning or alterations to training strategy, and can be applied to any off-the-shelf model and is compatible with both deterministic and stochastic samplers. We first validate our framework on toy 2D data, and then demonstrate its application for diffusion models trained across five disparate tasks – image generation, pose estimation, depth prediction, robot manipulation, and protein design. We find that across these tasks, our approach allows sampling from sharper (or flatter) distributions, yielding performance gains *e.g.,* depth prediction models benefit from sampling more likely depth estimates, whereas image generation models perform better when sampling a slightly flatter distribution.

## 1. Introduction

Score-based generative models, such as denoising diffusion (Ho et al., 2020) and flow matching (Lipman et al., 2023; Liu et al., 2023b), have become ubiquitous across AI applications. Given training data $\{\mathbf{x}^n\}$, they can model the underlying data distribution $p(\mathbf{x})$ (or $p(\mathbf{x}|\mathbf{c})$ for conditional settings) and at inference, they allow drawing samples

$\mathbf{x} \sim p(\mathbf{x})$ *e.g.,* to generate novel images. However, in certain applications, we may not want to truly sample the original data distribution. For example, when predicting depth from RGB input, we may want the more likely estimate(s) as output. In contrast, an artist exploring design choices may want the trained image generative model to yield more diverse samples, even if they may be somewhat less likely in the data. In this work, we ask whether we can steer the sampling process of diffusion or flow matching models to output more likely (or conversely, more diverse) samples than the original training data.

This process of trading off sample likelihood and diversity at inference is commonly referred to as *temperature sampling* (Hinton et al., 2015) – a higher temperature leads to diverse samples, and a lower temperature leads to more likely ones. While prior methods have investigated temperature sampling for score-based generative models like denoising diffusion, developing an efficient temperature sampling method for pre-trained diffusion/flow models remains an open challenge. For example, commonly leveraged techniques like classifier-free guidance (Ho & Salimans, 2022) or variance-reduced sampling (Yim et al., 2023; Geffner et al., 2025) can trade off sampling diversity and likelihood, but as we show later, these are not probabilistically interpretable as temperature scaling the data distribution. Conversely, methods such as likelihood-weighted finetuning (Shih et al., 2023) or Langevin correction (Song et al., 2021b; Du et al., 2023) can indeed allow temperature sampling, but at the cost of additional training or significantly increased inference-time computation. In this work, we instead seek to develop a (local) temperature sampling method that is: a) *training free i.e.,* does not require fine-tuning or distilling a pre-trained model, b) compatible with deterministic samplers *e.g.,* DDIM (Song et al., 2021a), c) efficient *i.e.,* does not increase the number of score evaluations at inference, and d) provably correct for some simple distributions.

Towards developing such an approach, we note that denoising diffusion and flow matching models define a forward process to induce noisy data distributions $p(\mathbf{x}_t)$ and train neural networks to approximate the corresponding score

---

[*]Equal contribution  [1]School of Computer Science, Carnegie Mellon University, Pittsburgh, USA [2]Department of Computer Science, Stanford University, California, USA. Correspondence to: Yanbo Xu <yanboxu@princeton.edu>.

*Proceedings of the 43$^{rd}$ International Conference on Machine Learning*, Seoul, South Korea. PMLR 306, 2026. Copyright 2026 by the author(s).

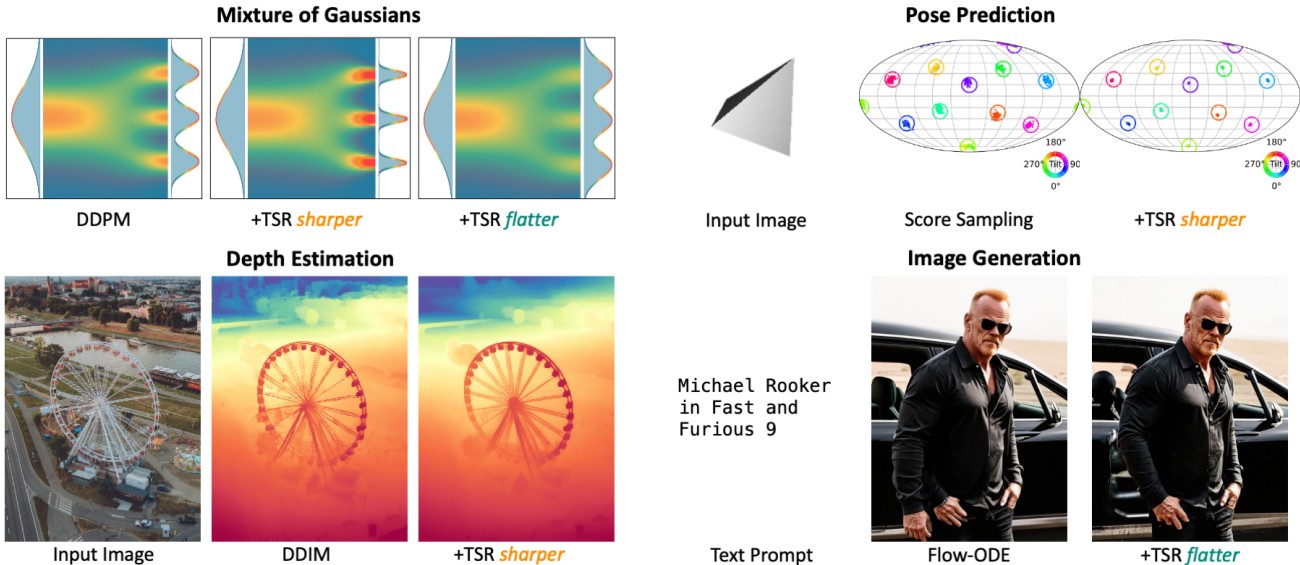

*Figure 1.* **Temporal Score Rescaling (`TSR`)** provides a mechanism to steer the sampling diversity of diffusion and flow models at inference. *Top-left:* Probability density evolution when sampling a 1D Gaussian mixture with DDPM, and the effects of `TSR`, which can control the sampling process to yield sharper or flatter distributions. *Top-right, bottom:* `TSR` can be applied to any pre-trained diffusion or flow model, improving performance across diverse domains such as pose prediction, depth estimation, and image generation.

functions $\nabla \log p(\mathbf{x}_t)$. We ask whether one can analytically relate these to the score of the (hypothetical) distributions $\bar{p}(\mathbf{x}_t)$ that would be induced by the forward process if the original data distribution were temperature scaled. We study the case of mixture of isotropic Gaussians, and derive a simple (time-dependent) rescaling function. As the reverse sampling process for sampling flow/diffusion models relies only on the learned score functions, our derived rescaling thus allows a training-free approach by simply scaling the inferred score at each inference step. While the analytical derivation is restricted to a simple setting, we show that our approach can be generally interpreted as a 'local' temperature sampling method, where it does not alter the overall distribution of global modes, but controls the local variance of samples around it.

We perform experiments to highlight the broad applicability of `TSR`. We show that it can efficiently allow local temperature sampling for denoising diffusion and flow matching models and is compatible with generic stochastic and deterministic samplers. We study diverse applications like image generation, depth estimation, pose prediction, robot manipulation, and protein generation. Across these applications, we show that `TSR` can provide a plug-and-play solution to control the sampling diversity of pre-trained models and leads to consistent performance gains *e.g.,* allowing more precise depth and pose inference, or enabling image generation to better match real data distribution.

## 2. Prior Art

**Guided Inference.** A widely adopted mechanism for steering sampling in diffusion and flow models is to leverage Classifier-Free Guidance (CFG) (Ho & Salimans, 2022). While this allows one to trade off likelihood and diversity by controlling the effect of the conditioning on the drawn samples, it is fundamentally different from temperature scaling. Moreover, CFG cannot be applied to unconditional models and even for conditional ones, requires training with condition dropout. An alternative to CFG by (Karras et al., 2024) is to use a 'bad version' of the diffusion model for guidance, but its probabilistic interpretation is unclear and it also requires intermediate checkpoints which are not widely available even for open-weight models. In comparison, `TSR` serves as a plug-and-play technique compatible with any diffusion and flow matching model without any requirement on training. Moreover, as we empirically demonstrate for image generation, our method is orthogonal to CFG and can be applied together for further improvement in quality.

**Temperature Scaling in Diffusion Models.** We are not the first to consider temperature sampling in context of diffusion models. In particular, (Shih et al., 2023) presented a technique to finetune diffusion (and autoregressive) models for temperature scaled inference. Their approach assigned an importance weight to each training sample based on its likelihood approximated by computing its ELBO with respect to a pretrained diffusion model on the same data. However, this approach is not training-free, making it difficult to leverage for large models and impossible in scenarios where training

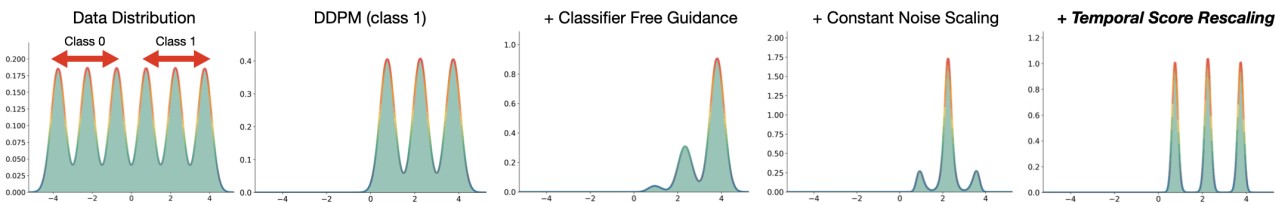

*Figure 2.* **Comparison on Uniform Mixture of 1D Isotropic Gaussians.** The uniform mixture of Gaussians distribution is divided into two classes (subplot 1). We apply CFG, CNS, and `TSR` to scale the conditional distribution of Class 1 (subplot 2). CFG and CNS lead non-uniform weights and tend to lose modes, while `TSR` preserve all modes and effectively reduce the variance of the samples.

data is unavailable. An alternative training-free approach is to modify the reverse sampling by applying stochastic MCMC corrector steps (Song et al., 2021b; Du et al., 2023), but can incur 5 times additional computation. Recent work (Skreta et al., 2025) avoid excessive corrector steps by re-weighting on batches of samples, which is inefficient when only one sample is desired (*e.g.,* image generation, robot manipulation). In contrast, `TSR` is a training-free approach that has *zero inference overhead*.

**Pesudo-temperature Sampling via Noise Scaling.** Perhaps the closest to our approach in terms of being efficient and training-free is the technique of 'Constant Noise Scaling' (CNS) where one scales the stochastic noise at each sampling step by a constant. More formally, following the definition by (Song et al., 2021b), CNS can be viewed as sampling the following reverse SDE:

$$d\mathbf{x} = [f(\mathbf{x}, t) - g(t)^2 \nabla \log p_t(\mathbf{x})]dt + \frac{g(t)}{\sqrt{k}} d\bar{\mathbf{w}} \quad (1)$$

where $f(\mathbf{x}, t)$, $g(t)$ denote the drift and diffusion coefficient, and $d\bar{\mathbf{w}}$ is a standard Wiener process. Compared to regular reverse diffusion SDE, the noise term is scaled by a constant $1/\sqrt{k}$. While CNS is the de facto approach to control sample variance in several domains (Yim et al., 2023; Geffner et al., 2025), as (Shih et al., 2023) point out, it is only a *'pseudo temperature'* sampling method. Intuitively, the noise-to-score ratio controls the strength of exploration versus converging to distribution modes during sampling. By scaling down this ratio by a constant, CNS over-suppresses exploration at high noise levels and under-suppresses it at low noise levels, leading to inadequate exploration of the data space when the model should recover global structure. We empirically show in section 3 that CNS behaves differently from temperature scaling and drop modes even for simple distributions. Moreover, CNS only applies to stochastic samplers and struggles with modern flow-matching models (see section 5.1). In contrast, we propose a time-dependent score scaling schedule that preserves the global structure of the sampled distribution and is compatible with both deterministic and stochastic samplers.

## 3. Analysis

To understand the behavior of `TSR`, we first empirically validate it on toy data and show it is more effective in scaling the variance of samples while preserving each local mode compared to existing approaches. Then, we analyze how the input parameters $(k, \sigma)$ control `TSR` and interpret their meanings in general settings.

### 3.1. Validation on Toy Distributions

**Mixture of 1D Gaussians.** We begin with a simple conditional generation task using a uniform mixture of 1D isotropic Gaussians in figure 2, where the left three and right three modes correspond to two different classes. We apply classifier-free guidance (CFG) with guidance scale 10, constant noise scaling (CNS) and `TSR` with $k = 10$ individually to scale the conditional distribution and evaluate whether each method preserves all modes under scaling. As shown in figure 2, CFG produces imbalanced samples, often favoring outer modes, while CNS shifts mass toward central modes at the expense of others. By contrast, `TSR` samples evenly across all modes while reducing intra-mode variance, demonstrating that it preserves the multimodal structure even under conditioning.

**General 2D Distributions.** We also apply `TSR` to unconditional generation on two complex 2D distributions: checkerboard and swiss roll. We train a small-scale diffusion model for each distribution and compare the scaled distribution sampled by CNS and `TSR` in figure 3. We observe that CNS consistently biases samples toward the central modes, resulting in mode collapse and poor coverage of peripheral regions. This supports the intuition that reducing noise too aggressively restricts exploration during the sampling process. In contrast, `TSR` maintains coverage of the global distribution while reducing local variance around each mode, producing samples aligning with the true distribution. These results show that, although derived for isotropic Gaussian data, `TSR` generalizes to more complex scenarios and provides consistent improvements in both conditional and unconditional generation.

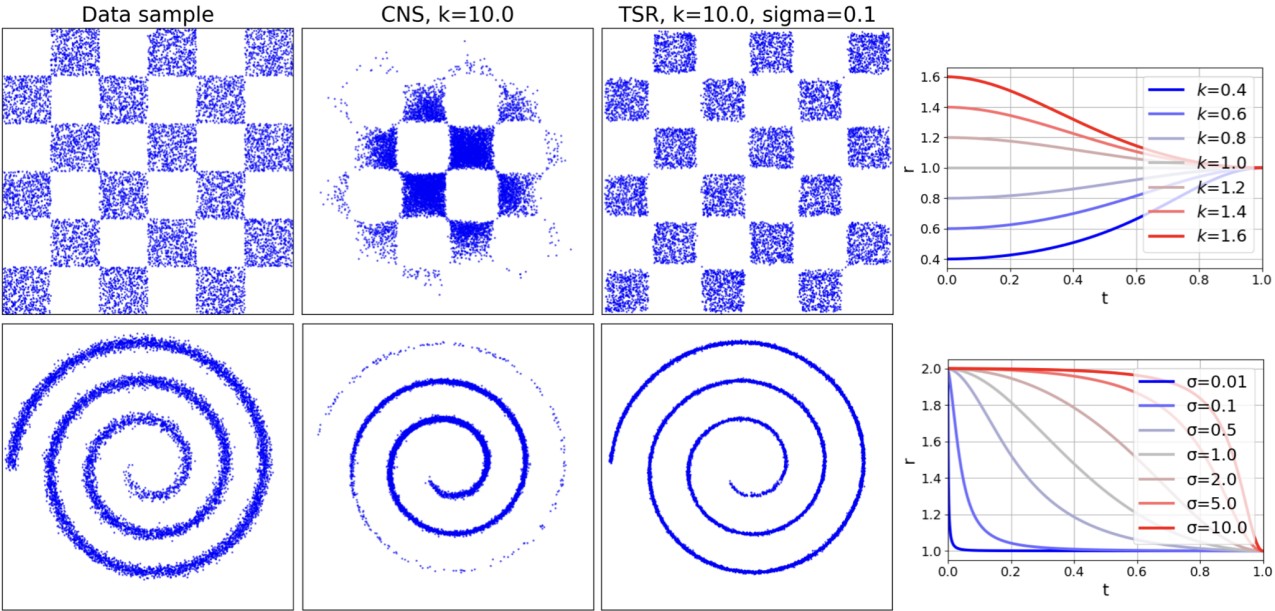

*Figure 3.* **Left: Comparison on 2D Checkerboard and Swiss Roll Distributions.** We compare samples from CNS and `TSR`. While CNS biases sampling towards the central modes and drops peripheral ones, `TSR` preserves all modes while reducing variance without generating divergent samples. **Right: Effect of Hyperparameters on the Rescaling Factor.** In the rightmost column, we plot the `TSR` rescaling factor $r_t$ on y-axis against diffusion time $t$. With $\sigma = 1.0$, varying $k$ controls the asymptotic value of $r_t$ (top); with $k = 2.0$, varying $\sigma$ determines how early rescaling takes effect during sampling (bottom).

### 3.2. Interpreting Rescaling Hyperparameters

In the derivation of `TSR`, $k$ referred to the factor of variance reduction and $\sigma$ referred to the variance of the modes in the data distribution. However, in real-world scenarios with more complex distributions, the variance of the data distribution is unknown. We provide an intuitive explanation of the role of $k$ and $\sigma$ on the rescaling factor $r_t$ to democratize the practical use of `TSR` in various scenarios. Specifically, we show how the rescaling factor $r_t$ changes over sampling time with different $k$ and $\sigma$ values in Fig. 3. Intuitively, $k$ indicates the max/min of the rescaling factor $r_t$. As $t \to 0$, signal-to-noise ratio $\eta_t \to \infty$, and $r_t \to k$. Meanwhile, $\sigma$ indicates how early we want to steer the sampling process. The larger $\sigma$, the earlier the sampling is steered. A very small $\sigma$ lets us use the original diffusion sampling ($r_t \approx 1.0$) and only steer the last few denoising steps.

## 4. Formulation
### 4.1. Preliminaries

Both diffusion and flow matching models fall under the family of stochastic interpolants (Albergo et al., 2023), which convert samples from data distribution $\mathbf{x}_0 \sim p_0(\mathbf{x})$ to gaussian noise $\epsilon \sim \mathcal{N}(0, \mathbf{I})$. The interpolant process can be defined as:

$$\mathbf{x}_t = \alpha_t \mathbf{x}_0 + \sigma_t \epsilon \qquad (2)$$

Different noise schedules $\alpha_t, \sigma_t$ correspond to different formulations of stochastic interpolants. For example, for flow

matching models, it is common to set $\alpha_t = 1 - t$, $\sigma_t = t$, while for variance-preserving diffusion models, they are defined such that $\alpha_t^2 + \sigma_t^2 = 1$.

We can sample from the data distribution by training a model $\mathbf{s}_\theta(\mathbf{x}, t) = \nabla \log p_t(\mathbf{x})$ that estimates the score of the noisy distribution. Starting from $\mathbf{x}_T \sim \mathcal{N}(0, \mathbf{I})$, the sampling process usually solves either a reverse-time SDE or a probability flow ODE. In practice, the learned model could predict various equivalent parameterization of the score, such as noise $\epsilon_\theta(\mathbf{x}_t, t)$ (common in denoising diffusion) or the probability flow velocity $\mathbf{v}_\theta(\mathbf{x}_t, t)$ (common in flow matching), which can all be expressed as linear combinations of score and $\mathbf{x}_t$ (See section 4.3).

### 4.2. Temporal Score Rescaling

Given a pre-training score function $\mathbf{s}_\theta$, we are interested in designing a temperature sampling process that does not require training or additional computation at inference. In particular, we propose a mechanism that achieves *local temperature scaling*, which steers the variance of the sampled distribution while preserving the global distribution structure (*e.g.,* without mode dropping). More formally, we define local temperature scaling as the task that takes in a data distribution $p_0(\mathbf{x})$ modeled as a mixture of (*an unknown set of*) Gaussians and generates the corresponding 'sharper' or 'flatter' distributions $\tilde{p}_0^k(\mathbf{x})$ (parameterized by

$k$):

$$p_0(\mathbf{x}) \equiv \sum_m w_m \mathcal{N}(\mathbf{x}; \mu_m, \Sigma_m)$$

$$\Rightarrow \tilde{p}_0^k(\mathbf{x}) \equiv \sum_m w_m \mathcal{N}(\mathbf{x}; \mu_m, \frac{1}{k}\Sigma_m)$$

Intuitively, $\tilde{p}_0^k(\mathbf{x})$ represents a distribution where the variance near each local mode in the data distribution is scaled by $\frac{1}{k}$, while preserving all the means and weights. Such a local scaling effect is different from the traditional temperature scaling that would change the weights of modes and alter the distribution structure. We now formulate our problem statement as: How can we alter the pretrained score function $\mathbf{s}_\theta$ so that a diffusion or flow sampler yields $\tilde{p}_0^k(\mathbf{x})$ instead of $p_0(\mathbf{x})$?

**Isotropic Gaussian Data.** To instantiate this, we start with a simple scenario where the data are drawn from a single isotropic Gaussian distribution $\mathbf{x_0} \sim \mathcal{N}(\mu, \sigma^2 \mathbf{I})$. The target is to sample from the locally scaled distribution $\tilde{p}_0^k(\mathbf{x}) \equiv \mathcal{N}(\mu, \frac{\sigma^2}{k}\mathbf{I})$. Under the stochastic interpolant process (equation 2), we define $p_t(\mathbf{x})$, $\tilde{p}_t^k(\mathbf{x})$ as the noisy distributions at time $t$ for the original and scaled data distribution, respectively. Since both the original and scaled data distributions are Gaussian, their corresponding noisy distribution can also be shown to be Gaussian:

$$p_t(\mathbf{x}) = \mathcal{N}(\alpha_t \mu, (\alpha_t^2 \sigma^2 + \sigma_t^2)\mathbf{I})$$

$$\tilde{p}_t^k(\mathbf{x}) = \mathcal{N}(\alpha_t \mu(\alpha_t^2 \frac{\sigma^2}{k} + \sigma_t^2)\mathbf{I})$$

(3)

Then, we can derive the corresponding score functions for the above distributions:

$$\nabla \log p_t(\mathbf{x}) = -\frac{\mathbf{x} - \alpha_t \mu}{\alpha_t^2 \sigma^2 + \sigma_t^2},$$

$$\nabla \log \tilde{p}_t^k(\mathbf{x}) = -\frac{\mathbf{x} - \alpha_t \mu}{\alpha_t^2 \frac{\sigma^2}{k} + \sigma_t^2}$$

(4)

Comparing the two score functions above, we observe that the score for the scaled distribution and the score for the original distribution follow a time-dependent linear relationship:

$$\nabla \log \tilde{p}_t^k(\mathbf{x}) = \frac{\eta_t \sigma^2 + 1}{\eta_t \frac{\sigma^2}{k} + 1} \nabla \log p_t(\mathbf{x}) \qquad (5)$$

where $\eta_t = \alpha_t^2 / \sigma_t^2$ is the signal-to-noise ratio. Note that $k = 1.0$ recovers the original score. Given a score estimator $\mathbf{s}_\theta(\mathbf{x}, t) = \nabla \log p_t(\mathbf{x})$, we can compute the score of $\tilde{p}_t^k$ with the above score rescaling equation and thus sample from $\tilde{p}_0^k$ from the same sampling process.

**Mixture of Gaussians.** We can show that the score ratio relationship (equation 5) is also a valid approximation if the data distribution is a mixture of *well-separated* isotropic Gaussians. In section B, we prove that the expected error between the score computed by equation 5 and the real score are bounded at all timestep $t$. On the high level, we derive an exponential bound for small $t$, where the modes are well-separated and only one Gaussian component dominates. For large $t$, we derive a polynomial bound based on the intuition that the distributions are similar to pure noise $\mathcal{N}(0, \mathbf{I})$. The error vanishes at both ends when $t$ converges to 0 or 1. The maximum error at any intermediate $t$ also converges to zero as the modes becoome more separated. We also empirically verify these results in section A.5

### 4.3. Steering Inference in Diffusion and Flow Matching

While the above analytical derivation for a score rescaling function focused on simple distributions, we empirically find that it can be applied across generic distributions and we operationalize equation 5 to define TSR sampling, a simple algorithm for steering sampling in diffusion and flow models:

---

**Sampling with Temporal Score Rescaling** TSR $(k, \sigma)$

Given a pre-trained score model $s_\theta$, TSR sampling substitutes its score prediction with:

$$\tilde{\mathbf{s}}_\theta(\mathbf{x}, t) = r_t(k, \sigma)\, \mathbf{s}_\theta(\mathbf{x}, t),$$

$$r_t(k, \sigma) := \frac{\eta_t \sigma^2 + 1}{\eta_t \frac{\sigma^2}{k} + 1} \qquad (6)$$

where $k, \sigma$ are user-defined parameters, and $\eta_t$ is the signal-to-noise ratio of the forward process.

---

This makes TSR a plug-and-play method compatible with any parameterization of $s_\theta$ and sampling algorithm, since conversions between score and model predictions are always linear and invertible.

**Denoising Diffusion**: These models are typically instantiated via neural networks $\epsilon_\theta$ that learn to predict the noise added. We can infer the predicted score from this noise via a simple linear relation $s_\theta(\mathbf{x}, t) = -\sigma_t^{-1}\epsilon_\theta(\mathbf{x}, t)$. We can thus perform TSR sampling in denoising diffusion models by simply using a rescaled noise prediction $\tilde{\epsilon}_\theta(\mathbf{x}, t)$ in any diffusion sampler (e.g., DDPM, DDIM):

$$\tilde{\boldsymbol{\epsilon}}_\theta(\mathbf{x}, t) = r_t(k, \sigma)\boldsymbol{\epsilon}_\theta(\mathbf{x}, t) \qquad (7)$$

**Flow Matching**: For flow matching models predicting the probability flow velocity $v_\theta(x, t)$, the corresponding score function can be computed by (Ma et al., 2024):

$$\mathbf{s}_\theta(\mathbf{x}, t) = -\frac{\alpha_t \boldsymbol{v}_\theta(\mathbf{x}, t) - \dot{\alpha}_t \mathbf{x}}{\sigma_t(\dot{\alpha}_t \sigma_t - \alpha_t \dot{\sigma}_t)} \qquad (8)$$

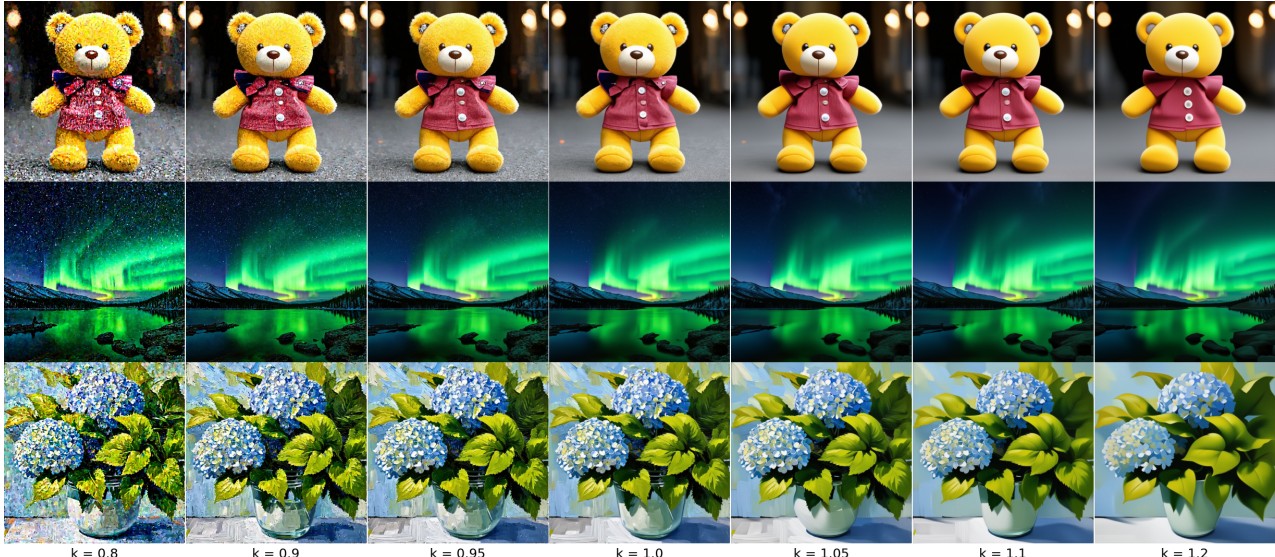

|  |  |  |  |  |  |  |
| k = 0.8 | k = 0.9 | k = 0.95 | k = 1.0 | k = 1.05 | k = 1.1 | k = 1.2 |

*Figure 4.* **Qualitative Examples for Varying** $k$**.** TSR allows for tuning the generated outputs to be more diverse and detailed (lower k) or more smooth and likely (higher k). While neither extreme is desirable, we notice a $k$ slightly smaller than 1 gives pleasing images with enhanced details.

Combining equation 6 and equation 8, we can derive the corresponding flow velocity $\tilde{\boldsymbol{v}}_\theta$ for the scaled distribution, such that $\tilde{\mathbf{s}}_\theta$ is a proper scaled version of the original score:

$$\tilde{\boldsymbol{v}}_\theta(\mathbf{x}, t) = \alpha_t^{-1}(r_t(k, \sigma)(\alpha_t \boldsymbol{v}_\theta(\mathbf{x}, t) - \dot{\alpha}_t \mathbf{x}) + \dot{\alpha}_t \mathbf{x}) \quad (9)$$

Applying this scaled velocity $\tilde{\boldsymbol{v}}_\theta$ in the flow samplers yields desired samples from the scaled distribution. Similar conversion can also be derived for other parameterizations of diffusion models like $x_0$-prediction and $v$-prediction.

## 5. Applications

We demonstrate the broad applicability and effectiveness of TSR by applying it to a diverse set of real-world applications, spanning image generation (section 5.1), protein design (section 5.2), depth estimation (section 5.3), pose prediction (section 5.4), and robot manipulation (section 5.5). For image generation, we find that a smaller $k$ enhances details and improves performance, while for other tasks, a larger $k$ yields higher accuracy of model predictions. To ease the parameter selection process, we include a general guideline in section 5.6.

### 5.1. Text-to-Image Generation

We examine the effect of steering the sampling distribution for diversity versus likelihood with TSR on Stable Diffusion 3 (Esser et al., 2024), a leading flow matching text-to-image model. As a creative task, image generation benefits from sampling a flatter distribution, which helps to recover more pleasing images with more high frequency details. We evaluate FID (Heusel et al., 2017; Parmar et al., 2022) and

CLIP (Radford et al., 2021) scores against a 5k image subset from LAION Aesthetics (Schuhmann et al., 2022) across different CFG guidance scale $w_{\text{cfg}}$, TSR parameter $k$ and $\sigma$. We fix the number of sampling steps to 30. In figure 5, we see adjusting $w_{\text{cfg}}$ makes a trade-off between text-alignment and image fidelity—higher $w_{\text{cfg}}$ increases CLIP score at the cost of worse FID. Meanwhile, TSR allows for additional improvement beyond the Pareto frontier of CFG. Compared to the regular Euler ODE sampling, TSR reduces *FID score from 24.77 ($\pm$ 0.10) to 22.81 ($\pm$ 0.13)* and increases *CLIP score from 32.82 ($\pm$ 0.014) to 33.05 ($\pm$ 0.018)*. These results are averaged over 5 random seeds. TSR achieves the optimal performance with $k = 0.93, \sigma = 3.0$. To verify the transferability of these parameters, we apply the same $(k, \sigma)$ to Flux.1 dev (Labs, 2024) and Stable Diffusion 2 (Rombach et al., 2022) and report the results in Table 1. The optimal $(k, \sigma)$ found on SD3 consistently improve performance on other models as well, suggesting the robustness of the choice for $(k, \sigma)$ across models.

Notably, while it is possible to perform stochastic sampling with flow models like SD3, we found that it performs significantly worse than ODE sampling with the same compute budget (see section A.1), making CNS impractical. We also show in section A.1 that TSR achieves superior performance with denoising diffusion model (SD2, (Rombach et al., 2022)) compared to CNS and other common samplers. Qualitatively, we observe in Fig. 4 that lower $k$ leads to images with more high-frequency detail (in the extreme case more noise), and higher $k$ leads to smoother images. We infer that using a smaller $k$ flattens the modeled distribution and allows better coverage of the desirable image

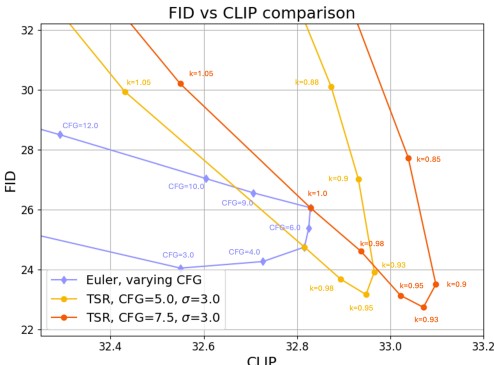

*Figure 5.* **Image Generation.** `TSR` achieves better text-alignment (CLIP) and image fidelity (FID), improving upon the Pareto frontier of CFG, which trades off between FID and CLIP.

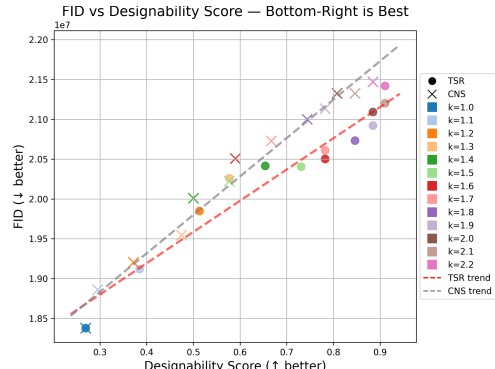

*Figure 6.* **Protein Generation.** `TSR` improves the designability score while preserving the diversity (FID) better compared to CNS. The original sampling has a designability score of 0.22.

|  | SD3 | | SD2 | | Flux.1 dev | |
|---|---|---|---|---|---|---|
|  | FID ↓ | CLIP ↑ | FID ↓ | CLIP ↑ | FID ↓ | CLIP ↑ |
| Default Scheduler | 24.77 | 32.82 | 22.81 | 33.66 | 53.99 | 31.97 |
| + TSR | **22.81** | **33.05** | **19.75** | **33.75** | **51.79** | **32.14** |

*Table 1.* **Evaluation of Text-to-Image Generation across Models**. `TSR` consistently improve image quality across Stable Diffusion 3 (Esser et al., 2024), Stable Diffusion 2 (Rombach et al., 2022), and Flux.1 dev (Labs, 2024). The optimal $(k, \sigma)$ found on SD3 generalize effectively to other models. For SD3 and Flux.1 dev, the default scheduler is Euler-ODE. For SD2 the default scheduler is DDPM.

|  | ETH3D | | NYUv2 | |
|---|---|---|---|---|
|  | AbsRel ↓ | $\delta_1$ ↑ | AbsRel ↓ | $\delta_1$ ↑ |
| DDIM | 7.1 | 90.4 | 6.0 | 95.9 |
| + CNS | 6.82 | 95.6 | 5.85 | **96.0** |
| +TSR | **6.68** | **95.7** | **5.84** | **96.0** |

*Table 2.* **Quantitative Evaluation of Depth Estimation.** `TSR` improves depth estimation and outperforms the naive baseline.

space. Overall, our results highlight that the control over the likelihood-diversity trade-off enabled by `TSR` is beneficial in image generation.

### 5.2. Protein Generation

Generative models have emerged as a powerful paradigm in AI for Science. For example, Protein discovery (Abramson et al., 2024; Jumper et al., 2021; Wu et al., 2024) is an application where such models have seen widespread adoption. However, not all generated proteins are valid in the real world. Thus, improving the designability of generated proteins is an important goal. CNS has previously been used to enhance sampling quality (Yim et al., 2023; Geffner et al., 2025).

We conduct experiments with FoldingDiff (Wu et al., 2024), a diffusion-based protein generation method, and compare `TSR` with CNS. Evaluation uses two complementary metrics: designability score(Wu et al., 2024), measuring structural quality and real-world feasibility, and protein FID(Faltings et al., 2025), capturing distributional similarity and thus diversity. Ideally, a method should achieve a high designability score and a low FID. As shown in Figure 6, samples from `TSR` lie in the bottom right regions, which shows `TSR` maintains protein diversity better than CNS, while improving the designability.

### 5.3. Depth Estimation

The task of monocular depth estimation is inherently challenging due to its uncertainty—an object may appear large but distant, or small but close. Recent methods (Duan et al., 2024; Saxena et al., 2023; Ke et al., 2024) address this with diffusion models, where different samples correspond to plausible variations or interpolations of the underlying depth structure. We adopt Marigold (Ke et al., 2024), which fine-tunes a pre-trained text-to-image diffusion model for depth estimation and achieves strong results. However, individual samples can be suboptimal due to both the sampling stochasticity and the ambiguity of depth estimation ( (Ke et al., 2024)). To mitigate this issue, it is desirable to increase the likelihood of each sampled estimate—i.e., to encourage samples to concentrate around the dominant modes of the learned distribution. Doing so reduces sampling variability and suppresses uncertain or noisy depth predictions.

We evaluate on the ETH3D (Schops et al., 2017) and NYUv2 (Nathan Silberman & Fergus, 2012) datasets. As shown in Table 2, `TSR` outperforms the default DDIM and CNS on prediction accuracy. By sampling from a sharper distribution, `TSR` yields more probable outputs given the input image. Qualitative comparisons in Fig.7 further show that `TSR` produces cleaner depth maps than DDIM, particularly in high-uncertainty regions.

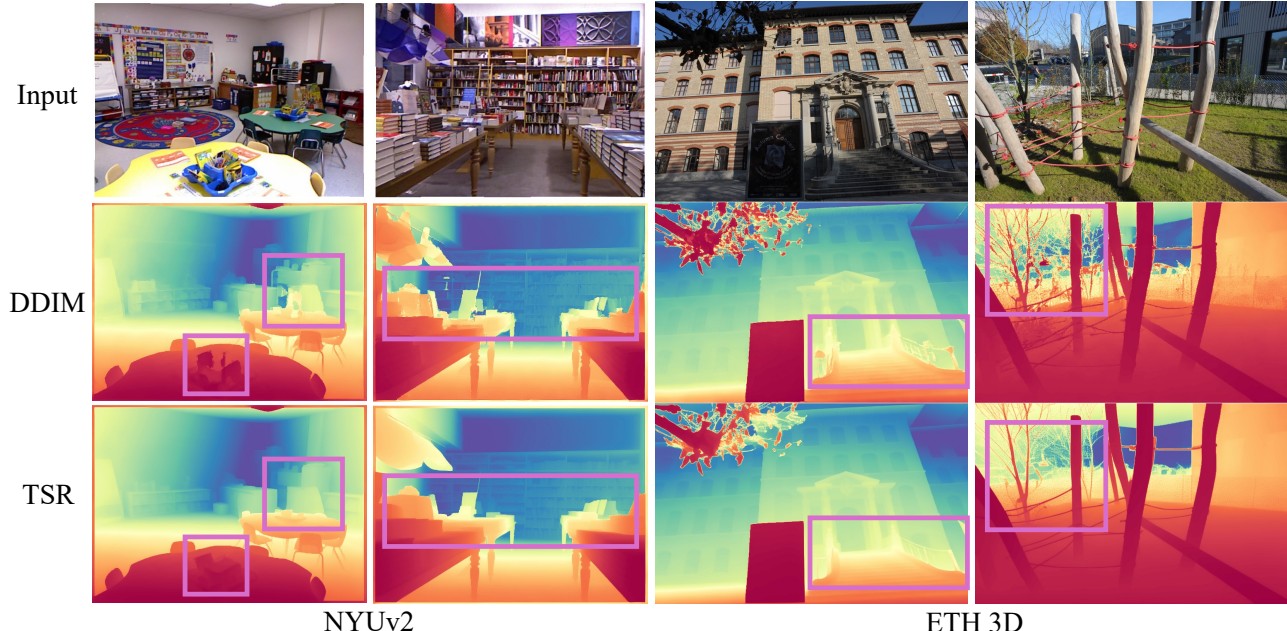

*Figure 7.* **Qualitative Depth Estimation Comparisons.** Compared to DDIM, TSR with $k > 1$ predicts cleaner depth in the regions with high uncertainty (highlighted by pink boxes).

|  | Error (deg) ↓ | Acc@ (deg) ↑ | | |
|---|---|---|---|---|
|  |  | 0.2 | 0.5 | 1.0 |
| Score Sampling | 0.444 | 9.4 | 68.3 | 97.9 |
| + CNS (1600) | **0.350** | **20.0** | **84.9** | **99.1** |
| + TSR (7.0, 0.5) | 0.356 | 18.5 | 84.0 | 99.0 |

*Table 3.* **Pose Prediction.** Mean error (deg) and accuracy within thresholds 0.2, 0.5, 1. $(k, \sigma) = (7.0, 0.5)$ for TSR, $k = 1600$ for CNS.

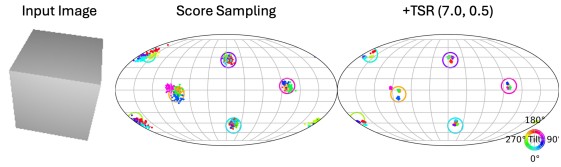

*Figure 8.* **Predicted poses on SYMSOL.** TSR reduces pose prediction error: each dot marks a sample's first canonical axis (colored by rotation), while circles denote ground-truth poses.

### 5.4. Pose Prediction

Previous work (Leach et al., 2022; Hsiao et al., 2024; Wang et al., 2023; Zhang et al., 2024) has shown that diffusion models can effectively predict object and camera poses in the $SO(3)$ space. We demonstrate TSR can improve such models' accuracy by sampling from a sharper distribution. Our evaluations are based on the $SO(3)$ diffusion models proposed by (Hsiao et al., 2024), where we apply TSR and evaluate on the SYMSOL dataset (Murphy et al., 2021), which contains geometric shapes with a high order of symmetries. We visualize the effect of TSR in 8 where we show the sampled poses on an example image from SYMSOL. TSR samples poses more concentrated around ground truth modes (the circle centers) than the baseline score matching sampling used in (Hsiao et al., 2024). In quantitative evaluation (3). TSR predictions have lower average error and higher accuracy under a range of accuracy thresholds compared to score matching sampling, highlighting the benefits of predicting samples close to modes. We find that CNS also reduces pose error, achieving a performance slightly better

than TSR on SYMSOL. However, we note that TSR remain robust and applicable over many tasks and sampling methods where constant noise scaling is not possible.

### 5.5. Robotic Manipulation

Lastly, we examine the applicability of TSR on predicting robot actions, with a focus on robotic manipulation. One notable difference of this domain compared to others is that the policy only models a distribution of actions for a short horizon, as it is a sequential decision-making problem.

We chose Pi-0 (Black et al., 2025), a generalist robotic flow-matching policy released by Physical Intelligence, finetuned on LIBERO (Liu et al., 2023a), a simulation benchmark for robotic manipulation. Specifically, we evaluate the policy over 10 tasks in the LIBERO-10 benchmark with shared $(k, \sigma)$ values. The results are in Table 4. Without any further training, TSR improves the performance of 6 tasks and maintains performance for 2 tasks. One notable point is that

| Model / Task ID | 0 | 1 | 2 | 3 | 4 | 5 | 6 | 7 | 8 | 9 | Average |
|---|---|---|---|---|---|---|---|---|---|---|---|
| Pi-0 | 86.0 | **97.3** | **80.7** | 96.0 | **84.7** | 93.3 | 81.3 | 94.7 | **23.3** | 80.0 | 81.7 |
| + TSR(1.25,0.25) | **86.7** | **97.3** | 77.3 | **97.3** | **84.7** | **96.0** | **82.7** | **96.0** | 21.3 | **88.7** | **82.8** |

*Table 4.* **Results for Robotic Manipulation.** Success rates are computed across 3 seeds, 10 tasks, with 50 rollouts per task. The results are computed with the best $(k, \sigma)$ for TSR.

the two tasks (Task ID 2 and 8) where TSR shows worse performance are precisely those in which the base Pi-0 policy itself exhibits low success rates. This suggests that the suboptimal performance may be due to a common 'sharpening' ($k > 1$) hyper-parameter across tasks as this may be suboptimal when the policy is not correct, and that tuning TSR's $k$ for each task may yield further gains.

### 5.6. Parameter Selection Guidelines

Similar to CFG (Ho & Salimans, 2022), the hyperparameters $\sigma$ and $\epsilon$ in TSR only need to be tuned once per model. Moreover, as shown in Table 1, the same set of hyperparameters can generalize across different models within the same task. To further simplify hyperparameter selection, we provide the following practical guidelines.

As shown in the appendix across multiple tasks (figure 9, figure 12, and figure 14), the effect of $\sigma$ and $\epsilon$ on generation quality follows a predictable parabola-like trend when varying one parameter while fixing the other. Based on this observation, we first fix $\sigma$ (e.g., $\sigma = 1.0$) and identify the optimal $\epsilon^*$ via binary search. We then fix $\epsilon^*$ and tune $\sigma$ in the same manner, optionally repeating the process once more if needed. This procedure is usually substantially more efficient than exhaustive grid search.

### 6. Discussion

We presented TSR, an approach to alter the sampling distribution for pre-trained diffusion and flow models. While we demonstrated its efficacy across several (toy and real) tasks, there are fundamental limitations worth highlighting. First, unlike temperature scaling, TSR can only alter the 'local' sampling and there might exist applications where a global temperature scaling is more desirable *e.g.,* TSR does not change the weights of the components in a gaussian mixture, only the variance. Moreover, while TSR does empirically steer the sampling diversity in generic scenarios, the theoretical guarantees are limited to simpler settings and one may be able to derive a better algorithm for different distributions. Nevertheless, as TSR can be readily applied to any off-the-shelf denoising diffusion and flow matching model, we believe it would a generally useful technique for the community to explore. In particular, the alternative strategy of 'constant noise scaling' is already adopted across applications (Yim et al., 2023; Geffner et al., 2025), and our work offers an alternative that is empirically better and more

widely applicable (*e.g.,* in deterministic sampling).

### Acknowledgments.

We thank Nicholas M. Boffi, Jun-Yan Zhu, and Gaurav Parmar for the insightful discussion and feedback. This work was supported in part by NSF Awards IIS-2345610 and IIS-2442282 and gifts from Google and CISCO.

### Impact Statement

This paper proposes an inference-time sampling technique for diffusion and flow models. When applied to off-the-shelf generative models, it may inherit any limitations or risks of the underlying model, such as the propagation of biases present in the training data.

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

# A. Additional Results

## A.1. Image Generation

We show additional qualitative examples generated by Stable Diffusion 3 with `TSR` in figure 11. We show `TSR` with varying $k$ and highlight the control over the expression of high-frequency details. To better demonstrate the effect of user-input parameters $(k, \sigma)$ on image quality, we plot FID and CLIP scores ablating over different $(k, \sigma)$ in figure 9. The trends in figure 9 clearly demonstrate that `TSR` with a $k$ slightly smaller than 1 and various $\sigma$ values improves both metrics. and the optimal performance is achieved at $k = 0.93, \sigma = 3.0$. We compare the regular Euler-ODE sampling and `TSR` on different CFG guidance scales in figure 10, highlighting that `TSR` is orthogonal to CFG and improves model performance at various CFG settings. In figure 10, we also present the performance of CNS, which has to be applied on a stochastic sampler (Euler-SDE). As Stable Diffusion 3 is a flow matching model, stochastic samplers perform significantly worse than ODE samplers, especially when the inference steps are less than 100. These results align with the findings in (Ma et al., 2024). Therefore, CNS does not practically apply to flow matching models like SD3. In Table 5, we additionally present quantitative results on Stable Diffusion 2, which is a denoising diffusion-based model. While CNS can improve over DDPM sampling, it is outperformed by `TSR`.

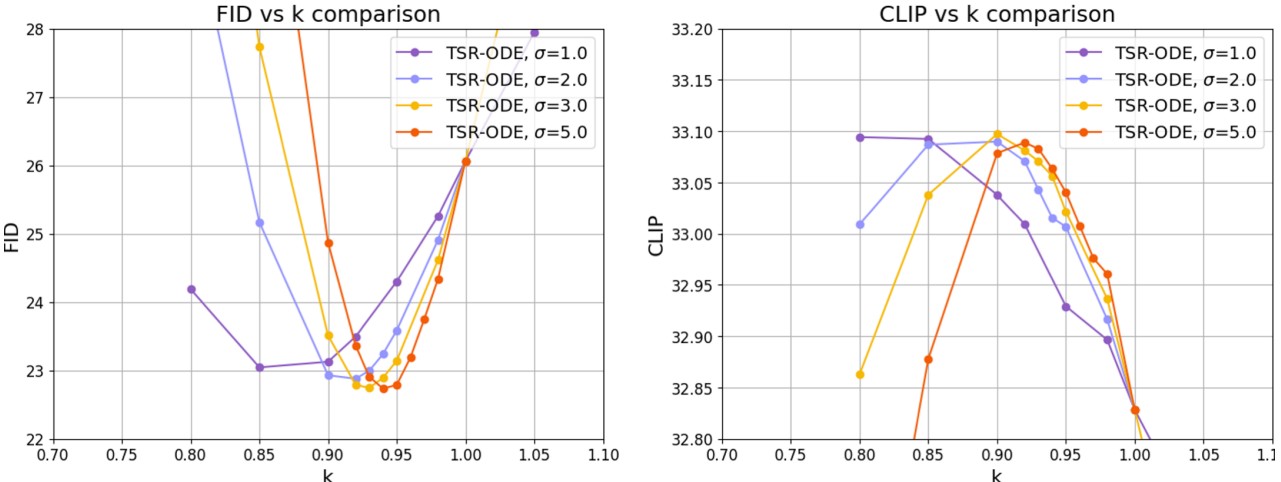

*Figure 9.* **Ablations over the `TSR` parameters** $(k, \sigma)$ **on Stable Diffusion 3**

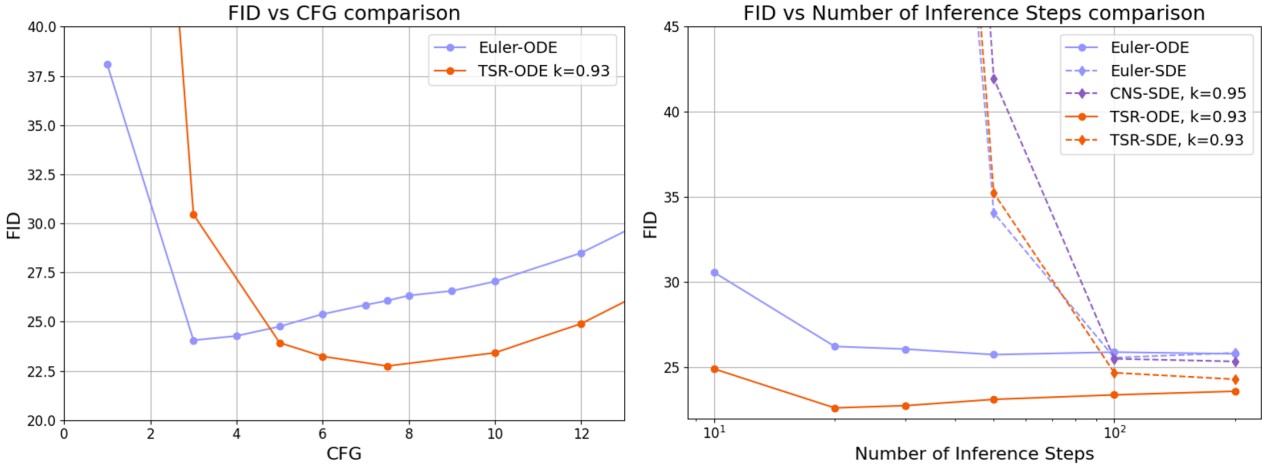

*Figure 10.* **Ablations over CFG scale and samplers** *Left*: Comparing regular sampling with `TSR` with various CFG guidance scale on Stable Diffusion 3. *Right*: Comparing deterministic and stochastic samplers with `TSR` or CNS. Stochastic sampling is much worse on flow models, making CNS impractical.

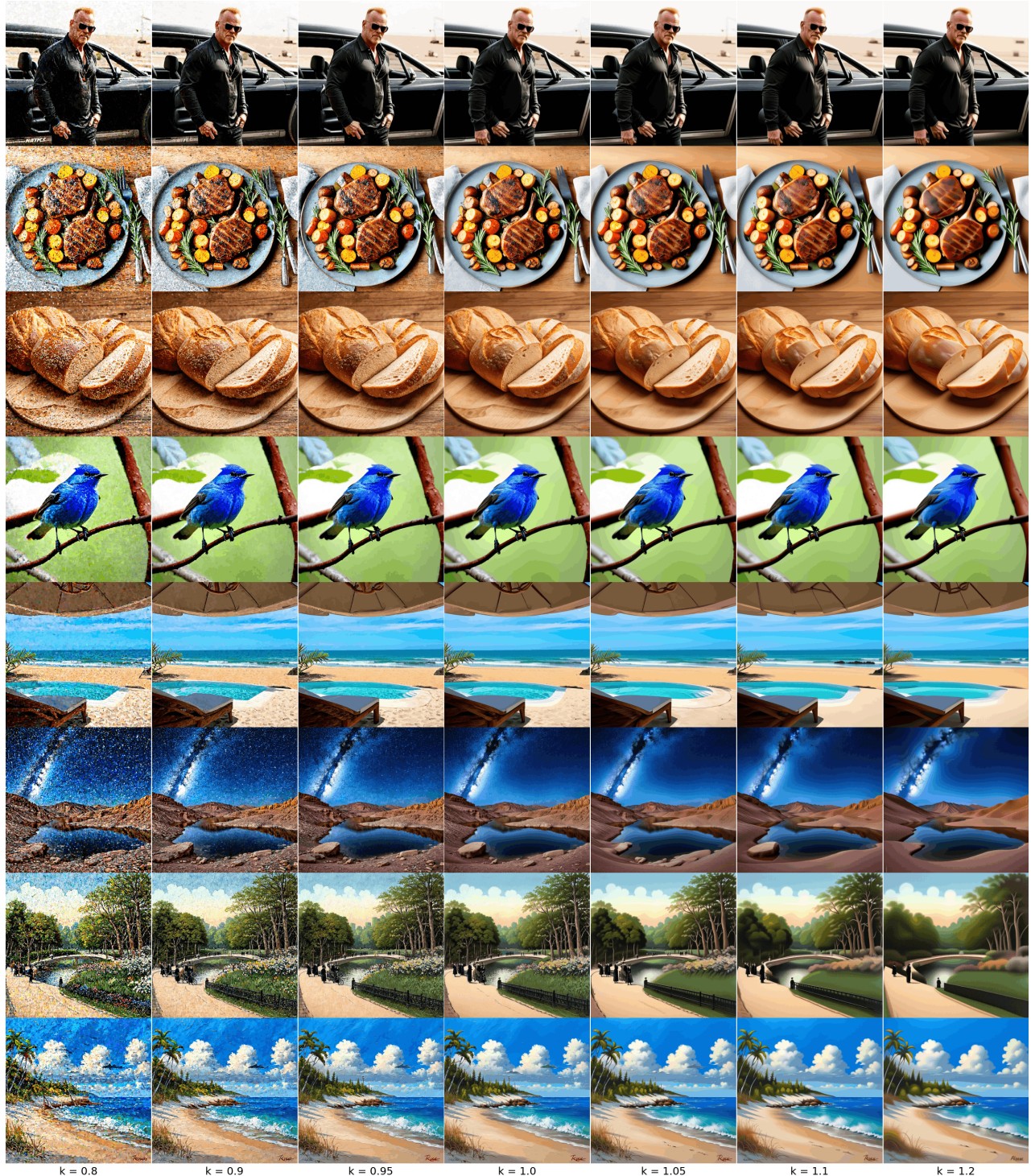

| k = 0.8 | k = 0.9 | k = 0.95 | k = 1.0 | k = 1.05 | k = 1.1 | k = 1.2 |

*Figure 11.* **Additional qualitative examples of TSR on Stable Diffusion 3**

|  | FID ↓ | CLIP ↑ |
|---|---|---|
| DDIM | 21.28 | 33.54 |
| + TSR (0.95, 1.0) | **20.05** | **33.61** |
| DDPM | 22.81 | 33.66 |
| + Constant Noise Scaling (0.96) | 19.87 | 33.68 |
| + TSR (0.9, 1.0) | **19.57** | **33.77** |
| EulerDiscrete | 22.11 | 33.54 |
| + TSR (0.95, 1.0) | **19.95** | **33.61** |

*Table 5.* **Results on Stable Diffusion 2.** TSR improve FID and CLIP on various samplers, outperforming CNS.

## A.2. Pose Prediction

We show more pose prediction results in figure 13. TSR predicts tighter samples around the ground truth mode, which can be observed by the low spread of sampled poses compared to score sampling. We also include ablations over parameters $(k, \sigma)$ in figure 12, showing that TSR consistently improves accuracy by increasing $k$, across different $\sigma$.

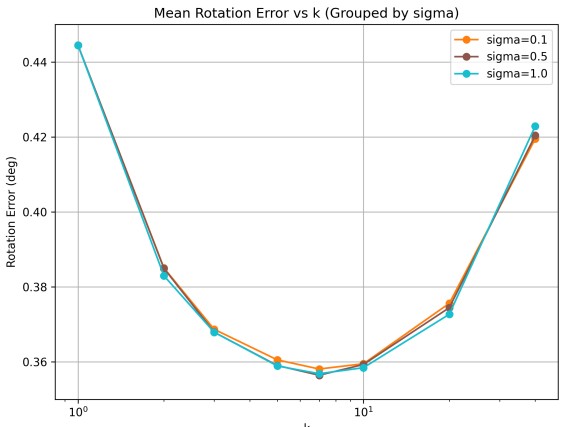

*Figure 12.* **Ablations over $(k, \sigma)$ on pose prediction.** TSR with various $(k, \sigma)$ configurations effectively outperforms the baseline sampling method $k = 1$. While TSR is not sensitive to $\sigma$ in pose estimation, TSR reaches optimal performance with $k \approx 7$.

## A.3. Depth Estimation

We also show the effect of $\sigma$ and $k$ on the AbsRel metric in Figure 14. Compared with the DDIM sample ($k = 1$), TSR demonstrates consistent performance gain in various $(k, \sigma)$ configurations. We also include more depth samples and comparisons Figure 15. A consistent improvement of TSR result can be observed, compared to the DDIM samples.

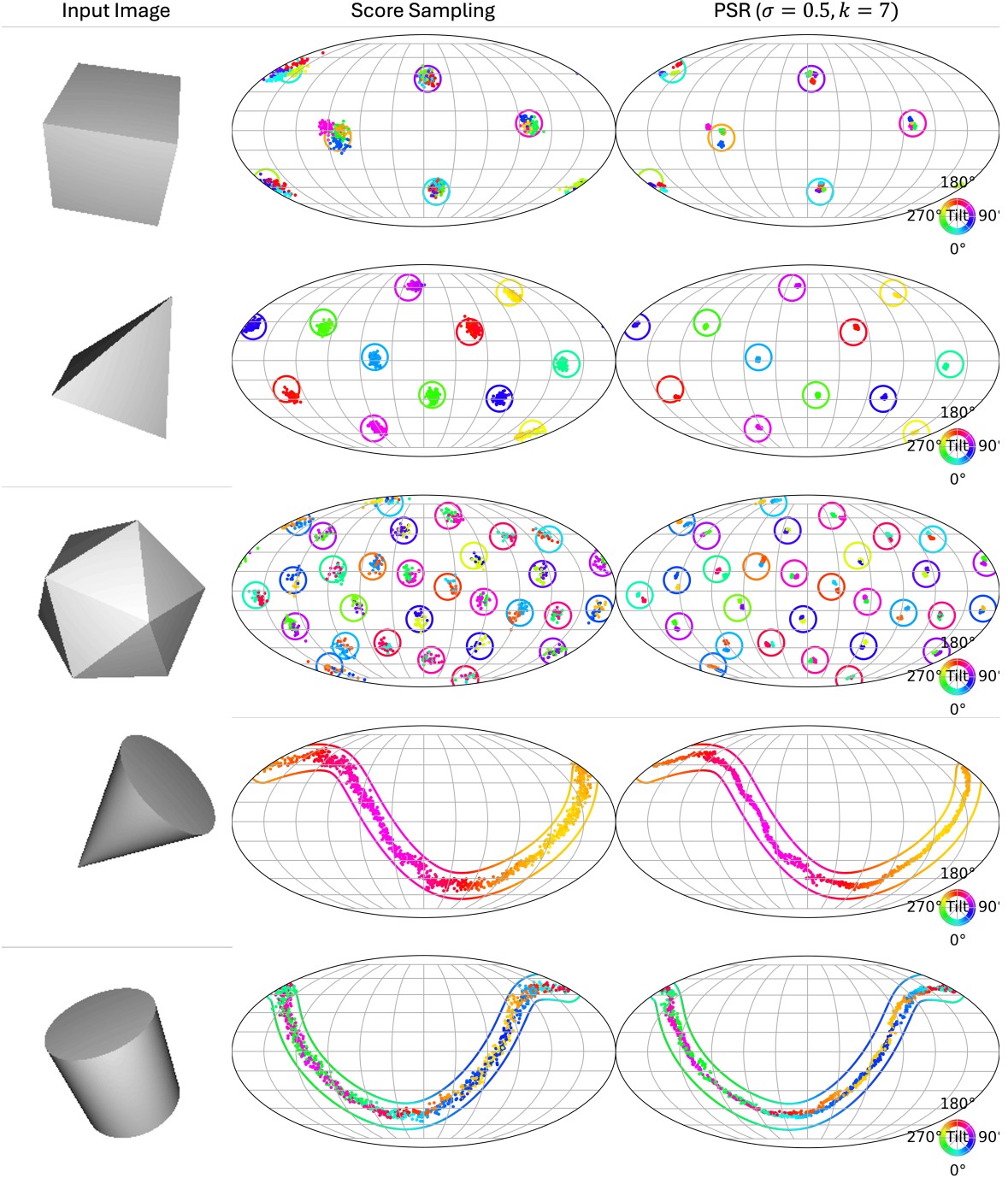

*Figure 13.* **More predicted poses on SYMSOL.** We show all 5 classes of shapes in SYMSOL. We use $\sigma = 1, k = 7$ for these visualizations. `TSR` consistently reduces prediction error across all classes compared to score sampling. We modify the location of samples to exaggerate error by a factor of 15 to show the visual difference given plotting constraints.

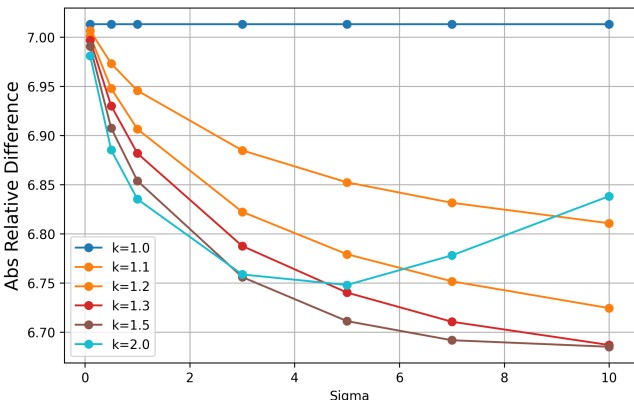

*Figure 14.* **Effects of** $(k, \sigma)$ **on depth estimation.** Comparing with DDIM sample ($k = 1$), TSR demonstrates consistent performance gains in various $(k, \sigma)$ configurations.

## A.4. Quantifying Mode Collapse

To systematically evaluate the mode-collapse behavior of temperature-scaling approaches (Constant Noise Scaling and TSR), we train an unconditional DDPM on the MNIST dataset( (Deng, 2012)) and apply each sampling method. We additionally train a classifier to label generated samples and assess whether Constant Noise Scaling or TSR exhibits mode drop, i.e., produces an imbalanced distribution of digits.

Figures 16 and 17 summarize the results, using $k = 5.0$ for Constant Noise Scaling (CNS) and $(k, \sigma) = (5.0, 1.0)$ for TSR. CNS disproportionately generates digits '1' (40.4%) and '9' (28.7%), likely because their straight or curved components appear frequently across other digits, making them easier to synthesize under decreased noise. In contrast, TSR produces a distribution of digits that closely matches that of DDPM, indicating that it preserves all modes. Furthermore, TSR generates noticeably clearer samples than DDPM, demonstrating the benefit of tempered sampling.

In summary, TSR maintains mode coverage on MNIST while improving sample quality.

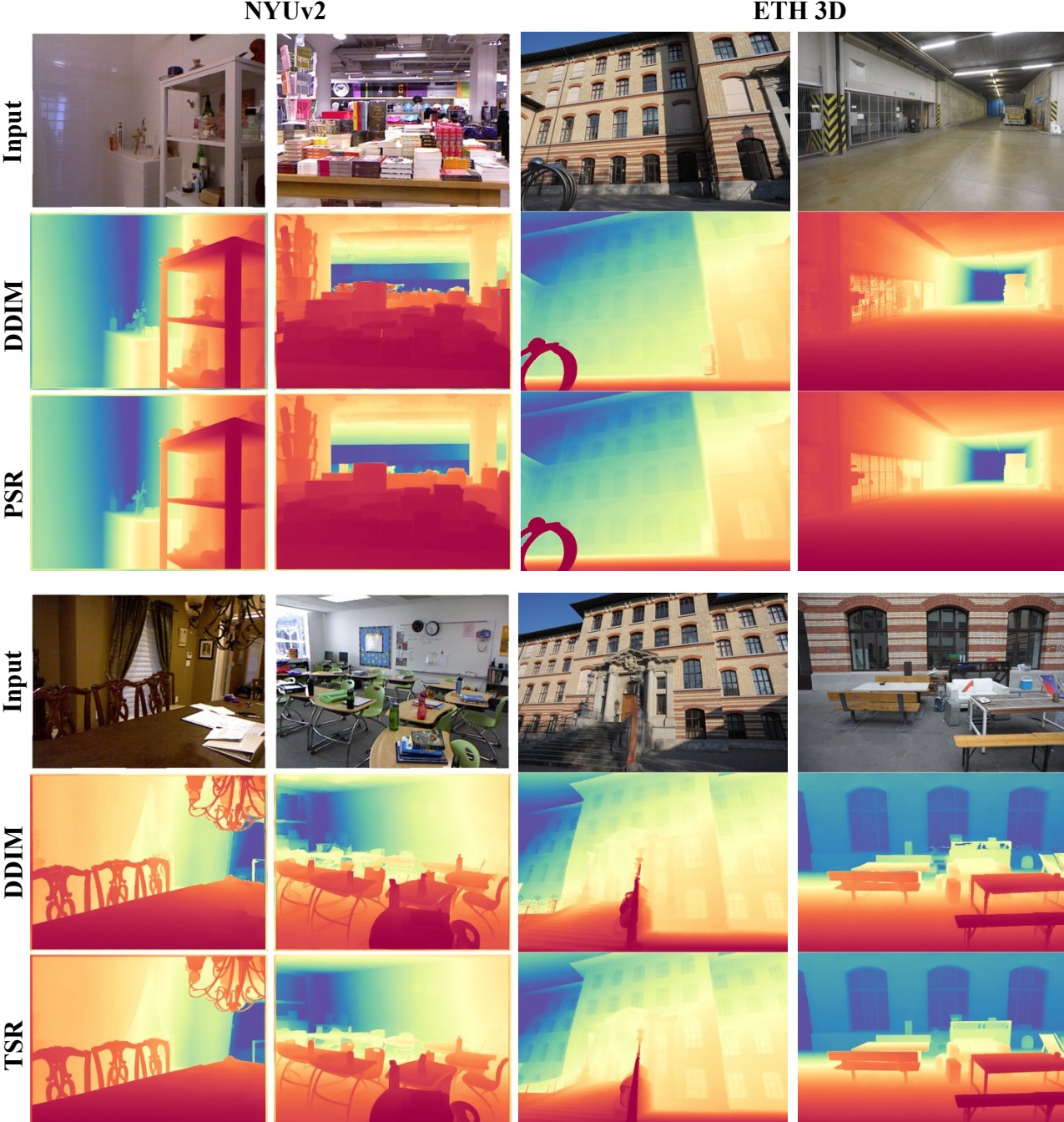

*Figure 15.* **More Depth Prediction Comparison.** We include more samples from NYUv2 and ETH3D. PSR demonstrates consistent improvement compared to the DDIM samples.

## A.5. Empirical Analysis on Score Approximation

To empirically analyze the our proposed approximation, we conduct an experiment using a 2D mixture of four Gaussian distributions, which is visualized in figure 18. We denote the distance between neighboring modes as $\Delta$ and the variance of each mode as $\sigma$. Setting the scaling parameter $k = 2$, we systematically vary $\Delta$ and $\sigma$ to study the behavior of the approximation error. We quantify the deviation by computing the expected absolute relative difference (Abs. Rel.) between the score estimated by TSR and the ground-truth score.

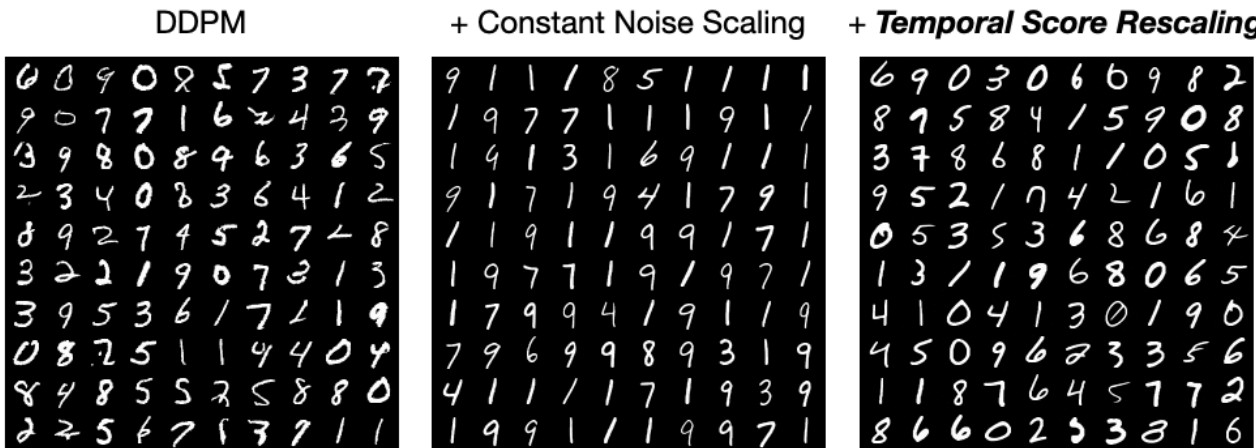

*Figure 16.* Samples generated on MNIST using DDPM, Constant Noise Scaling (CNS), and `TSR`. CNS tends to favor generating 1 and 9 while making `TSR` produces clearer digits while preserving diversity across modes.

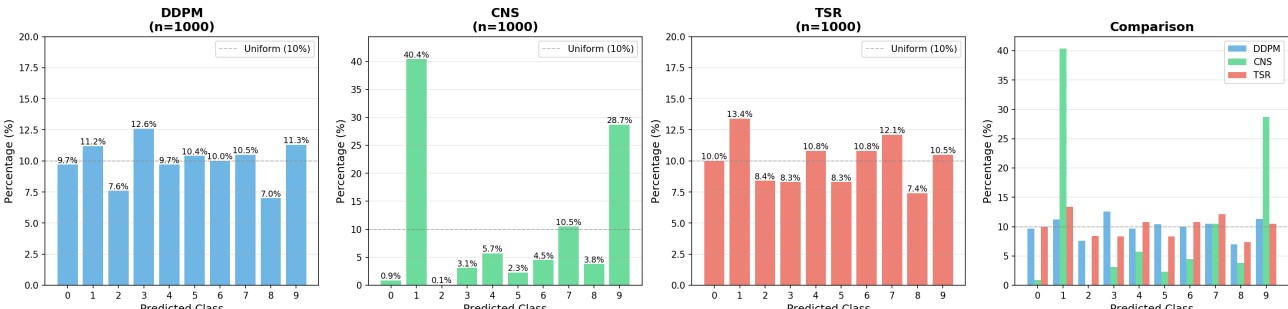

*Figure 17.* Class distribution of generated MNIST samples under DDPM, CNS, and `TSR` (CNS: $k = 5.0$; `TSR`: $(k, \sigma) = (5.0, 1.0)$). CNS exhibits mode imbalance, whereas `TSR` maintains a balanced distribution consistent with the dataset.

As illustrated in figure 18, the error vanishes at both ends in the range of timestep $t$ peaks at intermediate $t$. Furthermore, we analyze the maximum error occurring across all timesteps with respect to $\sigma$ and $\Delta$. The results demonstrate that the maximum error vanishes as the mode variance $\sigma$ decreases or the mode separation $\Delta$ increases, verifying that the approximation becomes exact as the modes are more separated. These results empirically confirm the theoretical bound we proved in section B.

### A.6. Additional Comparison on 2D Toy Distribution

We compare against two additional existing methods that can have similar effect as CNS in temperature sampling. One is constant score scaling (CSS), adopted by (Skreta et al., 2025). Instead of scaling down the noise term like CNS in equation 10, CSS constantly scale the score prediciton at each diffusion step, which is equivalent to solving the following reverse SDE:

$$d\mathbf{x} = [f(\mathbf{x}, t) - kg(t)^2 \nabla \log p_t(\mathbf{x})]dt + g(t)d\bar{\mathbf{w}} \tag{10}$$

The other method is the MCMC corrector studied in (Du et al., 2023; Song et al., 2021b). At each CNS sampling step, we apply additional $n$ stochastic MCMC corrector steps with the score prediction scaled by $k$, which approach the sampled distribution toward the target tempered distribution $p(x)^k$.

We compare `TSR`against the above two methods, along with CNS, DDPM sampling on the same checkerboard setting shown in Fig. 3. We present the qualitative comparison in Fig. Constant score scaling (CSS) demonstrates similar mode-dropping behavior as CNS. We apply MCMC corrector with CNS with 2x, 5x, and 10x more score evaluation compared to regular

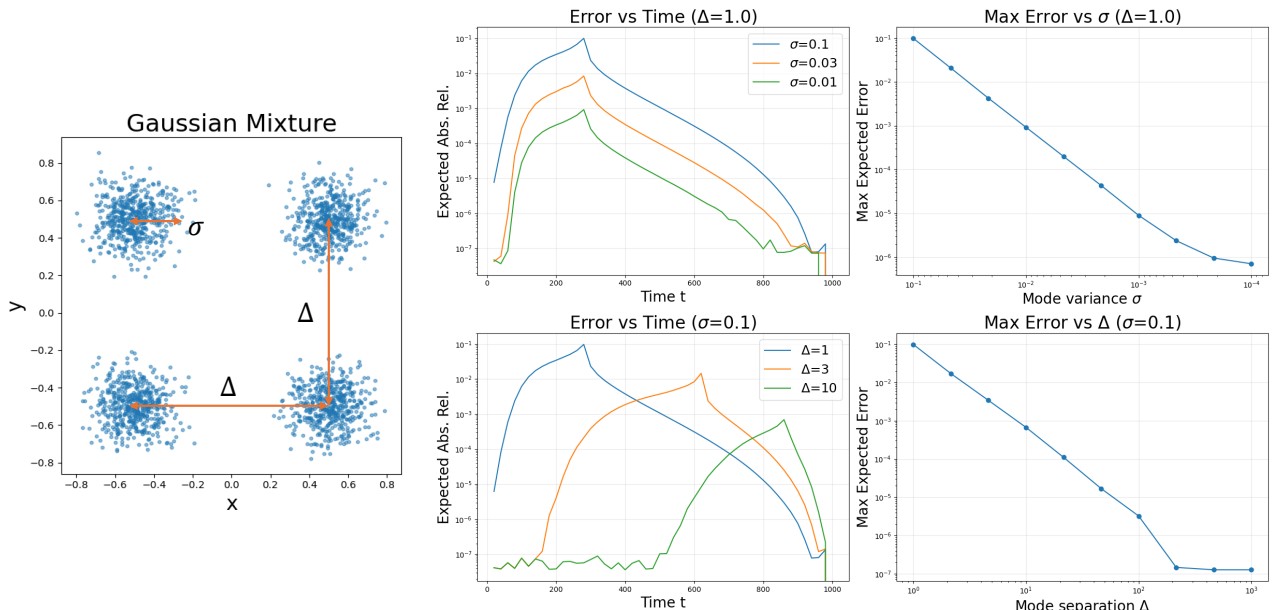

*Figure 18.* **Empirical Score Approximation Error**: For the mixture of gaussians depicted in the left, with mode distance $\Delta$ and mode variance $\sigma$, we compute the expected error of TSR approximation at $k = 2$. The maximum error is bounded and decreases as $\sigma$ decreases or $\sigma$ increases (right column).

sampling. While the MCMC corrector alleviates mode-dropping, it still prefer central modes. TSR is the only method that preserve the uniform weights across modes in this setting.

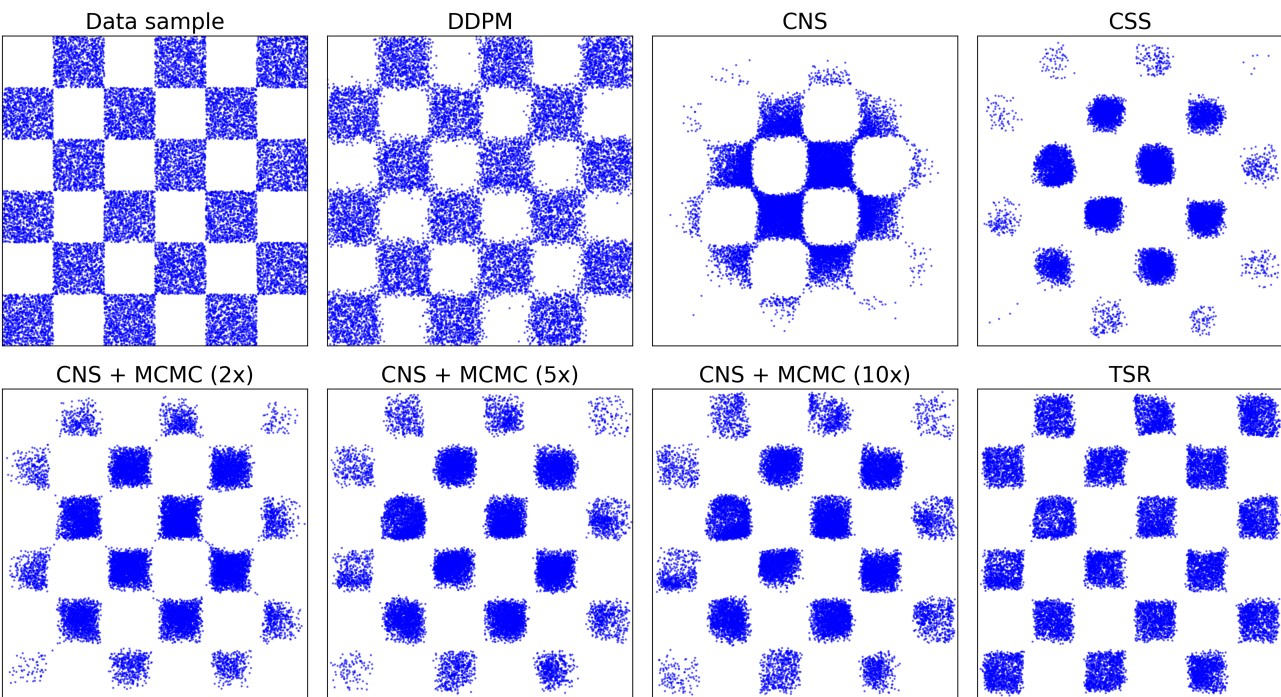

*Figure 19.* **Comparison with the CSS and MCMC baseline**: On the 2D checkerboard distribution, both CNS and CSS demonstrates mode-dropping behavior. We apply MCMC corrector to CNS with 2x, 5x, and 10x more score evaluations. While the MCMC corrector alleviates mode-dropping, it still prefer central modes.

# B. Proof for TSR for Mixture of well-separated Gaussians

We show that for a mixture of well-separated Gaussians, the score approximation in `TSR` is valid, with the approximation error vanishing asymptotically.

We begin by introducing the notation and defining the estimation error in section B.1. Our main result is stated in section B.2. The proof of this result is given in section B.3, supported by several lemmas whose proofs are provided in section B.4.

Notations:

- $\alpha_t, \sigma_t, k$: Diffusion/flow schedule coefficients and sharpening factor.

- $p_t^k(\mathbf{x})$: Induced distribution at time $t$ given the data distribution sharpened by $k$.

- $\Delta \gg \delta$: Distance between two mixture means at $t = 0$. Define $\Delta_t = \alpha_t \Delta$.

- $\sigma$: Variance of each Gaussian in the mixture at $t = 0$.

- $\sigma_{t,k}^2 \equiv \frac{\alpha_t^2 \sigma^2}{k} + \sigma_t^2$: Variance of each Gaussian at time $t$ with sharpening factor $k$.

- $\boldsymbol{\delta}_{t,n}(\mathbf{x}) \equiv \mathbf{x} - \alpha_t \boldsymbol{\mu}_n$: Offset vector from $\mathbf{x}$ to the center of the $n$-th Gaussian at diffusion time $t$.

- $p_{t,n}^k(\mathbf{x}) \propto \exp\left(-\frac{\|\boldsymbol{\delta}_{t,n}(\mathbf{x})\|^2}{\sigma_{t,k}^2}\right)$: Unnormalized density of $\mathbf{x}$ under the $n$-th Gaussian.

- $w_{t,n}^k(\mathbf{x}) \equiv \frac{p_{t,n}^k(\mathbf{x})}{\sum_m p_{t,m}^k(\mathbf{x})}$: Responsibility of the $n$-th Gaussian for $\mathbf{x}$.

- $N$: Number of Gaussians in the mixture. Dependent on the dataset only.

- $d$: Dimensionality of the data. i.e. $d = 2$ for 2D Gaussian Mixture.

- $\Delta_{\max} = \max_{i,j} |\boldsymbol{\mu}_i - \boldsymbol{\mu}_j|$: Maximum pairwise distance between Gaussian means in the mixture. For a general dataset, this term is bounded by $(N-1)\Delta$.

## B.1. Error in `TSR` Score Approximation.

**Score.** The score of the original data is given by:

$$\nabla \log p_t(\mathbf{x}) = -\frac{1}{(\alpha_t^2 \sigma^2 + \sigma_t^2)} \sum_n w_{t,n}^1(\mathbf{x}) \boldsymbol{\delta}_{t,n}(\mathbf{x})$$

For the target distribution $p^k(\mathbf{x}_0) = \sum_i \mathcal{N}(x; \boldsymbol{\mu}_i, \frac{\sigma^2}{k}\mathcal{I})$ the corresponding noisy distribution $p^k(\mathbf{x}_t) = \sum_i \mathcal{N}(x; \alpha_t \boldsymbol{\mu}_i, (\frac{\alpha_t^2 \sigma^2}{k} + \sigma_t^2)\mathcal{I})$, we have:

$$\nabla \log p_t^k(\mathbf{x}) = -\frac{1}{(\alpha_t^2 \sigma^2/k + \sigma_t^2)} \sum_n w_{t,n}^k(\mathbf{x}) \boldsymbol{\delta}_{t,n}(\mathbf{x})$$

In `TSR`, we approximate the score of $p^k(\mathbf{x}_t)$ by:

$$\nabla \log \tilde{p}_t^k(\mathbf{x}) \approx \frac{\alpha_t^2 \sigma^2 + \sigma_t^2}{\alpha_t^2 \sigma^2/k + \sigma_t^2} \nabla \log p_t(\mathbf{x}) = \frac{\sigma_{t,1}^2}{\sigma_{t,k}^2} \left(-\frac{1}{\sigma_{t,1}^2} \sum_n w_{t,n}^1 \boldsymbol{\delta}_{t,n}(\mathbf{x})\right)$$

$$= -\frac{1}{\sigma_{t,k}^2} \sum_n w_{t,n}^1 \boldsymbol{\delta}_{t,n}(\mathbf{x})$$

**Definition B.1** (Error in TSR Score Approximation). Define the amount of error in the score approximation as the expected difference between the scores:

$$Error(t) = \mathbb{E}_{\mathbf{x} \sim p_t^k} \frac{1}{\sigma_{t,k}^2} \| \sum_n (w_{t,n}^1(\mathbf{x}) - w_{t,n}^k(\mathbf{x})) \boldsymbol{\delta}_{t,n}(\mathbf{x}) \|$$

## B.2. Upper Bound of the Error

The objective of this proof is to establish a bound on the error term $Error(t)$. Our main results are as follows:

**Theorem B.2** (Upper Bound of the Error). *For $Error(t)$, there exists two upper bounds:*

$$Error(t) \leq B_{exp} = 6 \cdot \frac{\alpha_t \Delta_{\max}}{\sigma_{t,k}^2} \cdot \exp\left(-\frac{\alpha_t^2 \Delta^2}{8\sigma_{t,1}^2}\right)$$

$$Error(t) \leq B_{poly} = \frac{\alpha_t \Delta_{\max}}{4\sigma_{t,k}^2}\left(\frac{1}{\sigma_{t,k}^2} - \frac{1}{\sigma_{t,1}^2}\right) N\left(d\sigma_{t,k}^2 + \alpha_t^2 \Delta_{\max}^2\right)$$

**Theorem B.3** (Vanishing Behavior of Error). *Assuming $\sigma = \epsilon\Delta$, when $1 - \alpha_t^2 > \sqrt{\epsilon}$, we have:*

$$B_{\mathrm{poly}} \sim O(\sqrt{\epsilon})$$

*; When $1 - \alpha_t^2 \leq \sqrt{\epsilon}$, (i.e. $\alpha_t \approx 1$) we have:*

$$B_{\mathrm{exp}} \sim O(\frac{1}{\sqrt{\epsilon}\Delta} \ \exp(-\frac{1}{\sqrt{\epsilon}}))$$

**Conclusion.** Combining Theorem B.2 and Theorem B.3, when $\epsilon \to 0$, we have $Error(t) \to 0$. Therefore, when the Gaussians are well-seperated ($\epsilon \to 0$), the approximation error vanishes to 0.

## B.3. Proof of Theorem

Before proving the theorems, we first state several lemmas that are useful to the proof, whose proof will be given in the next section.

**Lemma B.4.** *The* `TSR` *approximation error $Error(t)$ is bounded as follows:*

$$Error(t) \leq \frac{\alpha_t \Delta_{max}}{\sigma_{t,k}^2} \mathbb{E}_{\mathbf{x} \sim p_t^k} \|dist(w_t^1(\mathbf{x}), w_t^k(\mathbf{x}))\| \tag{11}$$

*, where $dist(w_t^1(\mathbf{x}), w_t^k(\mathbf{x})) = \sum_n \|w_{t,n}^1(\mathbf{x}) - w_{t,n}^k(\mathbf{x})\|$.*

**Lemma B.5.** *There exists a polynomial bound for $\mathbb{E}_{\mathbf{x} \sim p_t^k}\|dist(w_t^1(\mathbf{x}), w_t^k(\mathbf{x}))\|$:*

$$\mathbb{E}_{\mathbf{x} \sim p_t^k} \|dist(w_t^1(\mathbf{x}), w_t^k(\mathbf{x}))\| \leq 6 \cdot \exp\left(-\frac{\alpha_t^2 \Delta^2}{8\sigma_{t,1}^2}\right)$$

**Lemma B.6.** *There exists an exponential bound for $\mathbb{E}_{\mathbf{x} \sim p_t^k}\|dist(w_t^1(\mathbf{x}), w_t^k(\mathbf{x}))\|$:*

$$\mathbb{E}_{\mathbf{x} \sim p_t^k} \|dist(w_t^1(\mathbf{x}), w_t^k(\mathbf{x}))\| \leq \frac{1}{4}\left(\frac{1}{\sigma_{t,k}^2} - \frac{1}{\sigma_{t,1}^2}\right) N\left(d\sigma_{t,k}^2 + \alpha_t^2 \Delta_{\max}^2\right)$$

*Proof of Theorem B.2.* Combining Lemma B.4 and Lemma B.5, we obtain the polynomial bound for $Error(t)$.

Similarly, Lemma B.4 and Lemma B.6 will give us the exponential bound for $Error(t)$.

$\square$

*Proof of Theorem B.3.* For simplicity, we assume diffusion scheduling, that is, $\sigma_t^2 = 1 - \alpha_t^2$ in this part. We also assume $\sigma = \epsilon\Delta$. As the dataset is fixed, we can rewrite $\Delta_{\max} = c\Delta$, where $c$ is a constant that only depends on the dataset.

### Vanishing of Polynomial Bound

Following the polynomial bound from B.2, we have:

$$B_{\mathrm{poly}} = N\frac{\alpha_t \Delta_{\max}}{\sigma_{t,k}^2}\left(\frac{(1 - 1/k)\sigma^2 \alpha_t^2}{\sigma_{t,1}^2 \sigma_{t,k}^2}\right)(d\sigma_{t,1}^2 + \alpha_t^2 \Delta_{\max}^2)$$

$$= N(1 - 1/k)\frac{\alpha_t^3 \Delta_{\max}\sigma^2}{\sigma_{t,k}^4}(d + \frac{\alpha_t^2 \Delta_{\max}^2}{\sigma_{t,1}^2})$$

Consider $1 - \alpha_t^2 > \sqrt{\epsilon}\Delta^2$, we have: $\sigma_{t,k}^2 = \alpha_t^2\sigma^2/k + (1 - \alpha_t^2) > (1 - \alpha_t^2) > \sqrt{\epsilon}\Delta^2$. Therefore, we have:

$$\frac{\alpha_t^3\Delta_{\max}\sigma^2}{\sigma_{t,k}^4}d \leq \frac{c\alpha_t^3\epsilon^2\Delta^3}{\epsilon\Delta^4}d = \alpha_t^3\,cd\,\frac{\epsilon}{\Delta}$$

Since $\alpha_t \leq 1$ and $c$ and $d$ are constant given a dataset, we can absorb them into a constant. Therefore, $\frac{\alpha_t^3 c\Delta_{\max}\sigma^2}{\sigma_{t,k}^4}d \leq C_1\frac{\epsilon}{\Delta}$, for some $C_1 = O(cd)$.

Similarly to previously proved, for the second term, $\frac{\alpha_t^3\Delta_{\max}\sigma^2}{\sigma_{t,k}^4} \cdot \frac{\alpha_t^2\Delta_{\max}^2}{\sigma_{t,1}^2}$, we have:

$$\frac{\alpha_t^5\sigma^2\Delta_{\max}^3}{\sigma_{t,k}^4\,\sigma_{t,1}^2} \leq \frac{\alpha_t^5 c^3\Delta^3(\epsilon^2\Delta^2)}{\epsilon\Delta^4 \cdot \sqrt{\epsilon}\Delta^2} \leq C_2\frac{\sqrt{\epsilon}}{\Delta}$$

, where $C_2$ is a constant term based on the dataset (and $\alpha_t$).

Therefore, we have the following.

$$B_{\text{poly}} \leq C_1\frac{\epsilon}{\Delta} + C_2\frac{\sqrt{\epsilon}}{\Delta} \leq C\frac{\sqrt{\epsilon}}{\Delta}$$

We can see that the polynomial bound is $O(\sqrt{\epsilon})$ for such $\alpha_t$, which goes to 0 as $\epsilon \to 0$

**Vanishing of Exponential Bound**

Assuming the diffusion schedule, and consider $\alpha_t$ such that $1 - \alpha_t^2 < \sqrt{\epsilon}\Delta^2$, we have:

$$B_{\exp} = 6\frac{\alpha_t\Delta_{\max}}{\alpha_t^2\sigma^2/k + 1 - \alpha_t^2}\ \exp\left(-\frac{\alpha_t^2\Delta^2}{8(\alpha_t^2\sigma^2 + 1 - \alpha_t^2)}\right)$$

With our assumption of $\sigma = \epsilon\Delta$, for a small $\epsilon$:

$$\begin{aligned}\alpha_{t,1}^2 = \alpha_t^2\sigma^2 + 1 - \alpha_t^2 &= \alpha_t^2\epsilon^2\Delta^2 + (1 - \alpha_t^2) \\ &\leq 2(1 - \alpha_t^2) \leq 2\sqrt{\epsilon}\,\Delta^2 \\ -\frac{\alpha_t^2\Delta^2}{8\alpha_{t,1}^2} \leq -\frac{\alpha_t^2\Delta^2}{8 \cdot 2\sqrt{\epsilon}\Delta^2} &= -\frac{\alpha_t^2}{16\sqrt{\epsilon}}\end{aligned}$$

Therefore,

$$\exp\left(-\frac{\alpha_t^2\Delta^2}{8\alpha_{t,1}^2}\right) \leq \exp\left(-\frac{\alpha_t^2}{16\sqrt{\epsilon}}\right)$$

As $\alpha_t^2\sigma^2/k + 1 - \alpha_t^2$ is dominant by $1 - \alpha_t^2$, we have $\alpha_t^2\sigma^2/k + 1 - \alpha_t^2 \approx 1 - \alpha_t^2$.

Therefore, we have:

$$B_{\exp} \leq 6\frac{\alpha_t\Delta_{\max}}{\alpha_t^2\sigma^2/k + 1 - \alpha_t^2}\ \exp\left(-\frac{\alpha_t^2}{16\sqrt{\epsilon}}\right) \approx \frac{6c\alpha_t}{\sqrt{\epsilon}\Delta}\exp\left(-\frac{\alpha_t^2}{16\sqrt{\epsilon}}\right)$$

As we consider $\alpha_t$ such that $1 - \alpha_t^2 < \sqrt{\epsilon}\Delta^2$, then we can write the exponential bound as $O(\frac{1}{\sqrt{\epsilon}\Delta}\ \exp(-\frac{1}{\sqrt{\epsilon}}))$, which also vanishes as $\epsilon \to 0$.

**Conclusion**

In both cases, at least one bound is vanishingly small as $\epsilon \to 0$.

$\square$

## B.4. Proof of Lemma

*Proof of Lemma B.4.* **Upper bound of the error**

Using the triangle inequality and the fact that $\sum_n w_{t,n}^1(\mathbf{x}) = 1$ and $\sum_n w_{t,n}^k(\mathbf{x}) = 1$, we have the following result:

$$
\begin{aligned}
Error(t) &= \mathbb{E}_{\mathbf{x}\sim p_t^k} \frac{1}{\sigma_{t,k}^2} \|\sum_n (w_{t,n}^1(\mathbf{x}) - w_{t,n}^k(\mathbf{x}))\boldsymbol{\delta}_{t,n}(\mathbf{x})\| \\
&\leq \frac{1}{\sigma_{t,k}^2} \mathbb{E}_{\mathbf{x}\sim p_t^k} \| \sum_n (w_{t,n}^1(\mathbf{x}) - w_{t,n}^k(\mathbf{x}))\|\boldsymbol{\delta}_{t,n}(\mathbf{x})\| \| \\
&\leq \frac{1}{\sigma_{t,k}^2} \mathbb{E}_{\mathbf{x}\sim p_t^k} \| \sum_n \left( (w_{t,n}^1(\mathbf{x}) - w_{t,n}^k(\mathbf{x}))\alpha_t \boldsymbol{\delta}_{\max} \right)\| \\
&\leq \frac{\alpha_t \boldsymbol{\delta}_{\max}}{\sigma_{t,k}^2} \mathbb{E}_{\mathbf{x}\sim p_t^k} \sum_n \|w_{t,n}^1(\mathbf{x}) - w_{t,n}^k(\mathbf{x})\|
\end{aligned}
$$

Therefore, the approximation error is bounded as follows:

$$
Error(t) \leq \frac{\alpha_t \Delta_{\max}}{\sigma_{t,k}^2} \mathbb{E}_{\mathbf{x}\sim p_t^k} \|dist(w_t^1(\mathbf{x}), w_t^k(\mathbf{x}))\| \tag{12}
$$

, where $dist(w_t^1(\mathbf{x}), w_t^k(\mathbf{x})) = \sum_n \|w_{t,n}^1(\mathbf{x}) - w_{t,n}^k(\mathbf{x})\|$.

$\square$

*Proof of Lemma B.5.* **Exponential Bound**

Following our problem setting, we have:

$$
p_t(\mathbf{x}) = \frac{1}{N} \sum_{i=1}^N \mathcal{N}(x; \alpha_t \boldsymbol{\mu}_i, (\alpha_t^2 \sigma^2 + \sigma_t^2)I).
$$

and

$$
q_t(\mathbf{x}) = \frac{1}{N} \sum_{i=1}^N \mathcal{N}(x; \alpha_t \boldsymbol{\mu}_i, (\frac{\alpha_t^2 \sigma^2}{k} + \sigma_t^2)I).
$$

, where $p_t(\mathbf{x})$ is the original distribution, and $q_t(\mathbf{x})$ is the desired distribution with altered variance.

For each x, the responsibility vector under a mixture is defined as:

$$
r^{(p)}(\mathbf{x}) = \left( r_1^{(p)}(\mathbf{x}), \dots, r_N^{(p)}(\mathbf{x}) \right)
$$

, where $r_i^{(p)}(\mathbf{x}) = \frac{\mathcal{N}(x; \alpha_t \boldsymbol{\mu}_i, \alpha_t^2 \sigma^2 + \sigma_t^2)}{\sum_{j=1}^N \mathcal{N}(x; \alpha_t \boldsymbol{\mu}_j, \alpha_t^2 \sigma^2 + \sigma_t^2)}$. $r^{(q)}(\mathbf{x})$ is defined analogously as $r_i^{(q)}(\mathbf{x}) = \frac{\mathcal{N}(x; \alpha_t \boldsymbol{\mu}_i, \alpha_t^2 \sigma^2/k + \sigma_t^2)}{\sum_{j=1}^N \mathcal{N}(x; \alpha_t \boldsymbol{\mu}_j, \alpha_t^2 \sigma^2/k + \sigma_t^2)}$.

Now we have $\mathbb{E}_{\mathbf{x}\sim p_t^k} \|dist(w_t^1(\mathbf{x}), w_t^k(\mathbf{x}))\| = \mathbb{E}_{\mathbf{x}\sim p_t^k}[D(\mathbf{x})]$, where $D(\mathbf{x}) := \|r^{(p)}(\mathbf{x}) - r^{(q)}(\mathbf{x})\|_1$.

Define $i(\mathbf{x}) = \max_i r_i$, and $e_i$ as the one-hot vector where the ith entry is one. Using the triangle inequality, we have:

$$
D(\mathbf{x}) = \|r^{(p)} - r^{(q)}\|_1 \leq \|r^{(p)} - e_{i_p(\mathbf{x})}\|_1 + \|e_{i_p(\mathbf{x})} - e_{i_q(\mathbf{x})}\|_1 + \|e_{i_q(\mathbf{x})} - r^{(q)}\|_1.
$$

, and that

$$
\begin{aligned}
\|r^{(p)} - e_{i_p(\mathbf{x})}\|_1 &= 2(1 - r_{i_p(\mathbf{x})}^p(\mathbf{x})) \\
\|e_{i_p} - e_{i_q}\|_1 &= 2 * \mathbf{1}\{i_p \neq i_q\} \\
\|r^{(q)} - e_{i_q(\mathbf{x})}\|_1 &= 2(1 - r_{i_q(\mathbf{x})}^q(\mathbf{x}))
\end{aligned}
$$

**Concentration of responsibilities for the true component** Let:

$$\epsilon := \max_{i \neq j} \mathbb{P}_{x \sim \mathcal{N}(\boldsymbol{\mu}_i, \sigma^2)} \left[ \|x - \boldsymbol{\mu}_j\| < \|x - \boldsymbol{\mu}_i\| \right]$$

That is, the probability that a sample from component $i$ is closer to another component $j$. Then:

$$\mathbb{E}_{x \sim p} \left[ 1 - \max_j r_j^{(p)}(\mathbf{x}) \right] \leq \epsilon \quad \Rightarrow \quad \mathbb{E}_{x \sim p}[D(\mathbf{x})] \approx 2\epsilon$$

Recall $\Delta := \min_{i \neq j} \|\boldsymbol{\mu}_i - \boldsymbol{\mu}_j\|$ to be the minimum pairwise distance between the means. Using Gaussian tail bounds, we can approximate:

$$\epsilon \approx \exp\left( -\frac{\Delta^2}{8\sigma^2} \right)$$

Hence, we have:

$$E_{x \sim p_t^k} \left( 2(1 - r_{i_p(\mathbf{x})}^p(\mathbf{x})) \right) \leq 2 \cdot \exp\left( -\frac{\alpha_t^2 \Delta^2}{8\sigma_{t,1}^2} \right)$$

$$E_{x \sim p_t^k} \left( 2(1 - r_{i_q(\mathbf{x})}^q(\mathbf{x})) \right) \leq 2 \cdot \exp\left( -\frac{\alpha_t^2 \Delta^2}{8\sigma_{t,k}^2} \right)$$

**Bounding** $\Pr(i_p \neq i_q)$

As $p_t(\mathbf{x})$ and $q_t(\mathbf{x})$ share the same modes, we have $\Pr(i_p \neq i_q) \leq \sum_i \Pr\left( i_p \neq i_q \mid x \sim \text{component } i \right) \Pr(x \text{ from } i)$, which can also be bounded using Gaussian tail bounds as above.

Therefore, we have:

$$\begin{aligned}
E_{x \sim p_t^k}(D(\mathbf{x})) &\leq E_x(\|r^{(p)} - e_{i_p(\mathbf{x})}\|_1) + E_x(\|e_{i_p(\mathbf{x})} - e_{i_q(\mathbf{x})}\|_1) + E_x(\|e_{i_q(\mathbf{x})} - r^{(q)}\|_1) \\
&= E_x\left( 2(1 - r_{i_p(\mathbf{x})}^p(\mathbf{x})) \right) + E_x\left( 2 * \mathbf{1}\{i_p \neq i_q\} \right) + E_x\left( 2(1 - r_{i_q(\mathbf{x})}^q(\mathbf{x})) \right) \\
&\leq 2 \cdot \exp\left( -\frac{\alpha_t^2 \Delta^2}{8\sigma_{t,1}^2} \right) + \left( \exp\left( -\frac{\alpha_t^2 \Delta^2}{8\sigma_{t,1}^2} \right) + \exp\left( -\frac{\alpha_t^2 \Delta^2}{8\sigma_{t,k}^2} \right) \right) + 2 \cdot \exp\left( -\frac{\alpha_t^2 \Delta^2}{8\sigma_{t,k}^2} \right) \\
&\leq 6 \cdot \exp\left( -\frac{\alpha_t^2 \Delta^2}{8\sigma_{t,1}^2} \right)
\end{aligned}$$

Finally:

$$E_{x \sim p_t^k}(D(\mathbf{x})) \leq 6 \cdot \exp\left( -\frac{\alpha_t^2 \Delta^2}{8\sigma_{t,1}^2} \right)$$

$\square$

*Proof of Lemma B.6.* **Polynomial Bound**

We consider the softmax representation of the responsibilities:

$$w_t^k(\mathbf{x}) = \text{softmax}\left( z_t^k(\mathbf{x}) \right), \quad \text{where} \quad z_{t,n}^k(\mathbf{x}) := -\frac{\|\mathbf{x} - \alpha_t \boldsymbol{\mu}_n\|^2}{2\sigma_{t,k}^2}.$$

. Using the Softmax Lipschitz bound that $\boxed{\| \text{softmax}(z) - \text{softmax}(z') \|_1 \leq 1/2 \|z - z'\|_1}$, we have:

$$\|w_t^k(\mathbf{x}) - w_t^1(\mathbf{x})\|_1 \leq \frac{1}{2} \|z_t^k(\mathbf{x}) - z_t^1(\mathbf{x})\|_1.$$

Compute the logits difference coordinatewise:

$$z_{t,n}^k(\mathbf{x}) - z_{t,n}^1(\mathbf{x}) = -\frac{\|\boldsymbol{\delta}_{t,n}(\mathbf{x})\|^2}{2\sigma_{t,k}^2} + \frac{\|\boldsymbol{\delta}_{t,n}(\mathbf{x})\|^2}{2\sigma_{t,1}^2}$$

$$= \frac{1}{2}\Big(\frac{1}{\sigma_{t,1}^2} - \frac{1}{\sigma_{t,k}^2}\Big)\|\boldsymbol{\delta}_{t,n}(\mathbf{x})\|^2.$$

Adding absolute values,

$$\|z_t^k(\mathbf{x}) - z_t^1(\mathbf{x})\|_1 = \frac{1}{4}\Big(\frac{1}{\sigma_{t,k}^2} - \frac{1}{\sigma_{t,1}^2}\Big)\sum_{n=1}^N \|\boldsymbol{\delta}_{t,n}(\mathbf{x})\|^2$$

**Bounding** $\mathbb{E}_x\Big[\sum_{n=1}^N \|\boldsymbol{\delta}_{t,n}(\mathbf{x})\|^2\Big]$

Let $x \sim p_t^k$ be drawn from the mixture with means $\{\alpha_t\boldsymbol{\mu}_i\}$ and variance $\sigma_{t,k}^2$. Write expectation as mixture-average:

$$\mathbb{E}_x\Big[\sum_{n=1}^N \|\boldsymbol{\delta}_{t,n}(\mathbf{x})\|^2\Big] = \frac{1}{N}\sum_{i=1}^N \mathbb{E}_{x\sim\mathcal{N}(\alpha_t\boldsymbol{\mu}_i,\sigma_{t,k}^2 I)}\Big[\sum_{n=1}^N \|x - \alpha_t\boldsymbol{\mu}_n\|^2\Big].$$

When the sample was generated from component i, for any other n, we have

$$\mathbb{E}\|x - \alpha_t\boldsymbol{\mu}_n\|^2 = \mathbb{E}\big[\|x - \alpha_t\boldsymbol{\mu}_i + \alpha_t\boldsymbol{\mu}_i - \alpha_t\boldsymbol{\mu}_n\|^2\big] = \mathbb{E}\|x - \alpha_t\boldsymbol{\mu}_i\|^2 + \|\alpha_t\boldsymbol{\mu}_i - \alpha_t\boldsymbol{\mu}_n\|^2$$

, because the cross-term has zero mean.

Since the first term equals the trace of the covariance $= d\sigma_{t,1}^2$, we have:

$$\mathbb{E}\|x - \alpha_t\boldsymbol{\mu}_n\|^2 = d\sigma_{t,1}^2 + \|\alpha_t(\boldsymbol{\mu}_i - \boldsymbol{\mu}_n)\|^2$$

Summing over all $N$ (including n=i, for which the pairwise term is zero) gives $\mathbb{E}_{x\sim\mathcal{N}(\alpha_t\boldsymbol{\mu}_i,\sigma_{t,1}^2 I)}\Big[\sum_{n=1}^N \boldsymbol{\delta}_{t,n}(\mathbf{x})\Big] = Nd\sigma_{t,1}^2 + \sum_{n=1}^N \|\alpha_t(\boldsymbol{\mu}_i - \boldsymbol{\mu}_n)\|^2$.

Now, bound the pairwise squared distances by the diameter squared: $\|\alpha_t(\boldsymbol{\mu}_{t,i} - \boldsymbol{\mu}_{t,n})\|^2 \le \alpha_t^2 \Delta_{\max}^2$.

Therefore, we have: $\mathbb{E}_x\Big[\sum_{n=1}^N \|\boldsymbol{\delta}_{t,n}(\mathbf{x})\|^2\Big] \le N\big(d\sigma_{t,1}^2 + \alpha_t^2\Delta_{\max}^2\big)$.

We then have the polynomial bound as:

$$\mathbb{E}_{x\sim p}[D(\mathbf{x})] \le \frac{1}{4}\Big(\frac{1}{\sigma_{t,k}^2} - \frac{1}{\sigma_{t,1}^2}\Big)N\big(d\sigma_{t,k}^2 + \alpha_t^2\Delta_{\max}^2\big)$$

$\square$

# C. Constant Noise Scaling

In this section, we provide a more detailed analysis of Constant Noise Scaling. As discussed in section **??**, CNS has been adopted as a practical technique to control trade-off sample variance and diversity. We intuitively explain and empirically verify that CNS does not correspond to true temperature scaling. We now provide a more rigorous proof that CNS cannot produce the temperature-scaled distribution. Following (Song et al., 2021b), a regular score-based model $\mathbf{s}_\theta(\mathbf{x}, t) = \nabla \log p_t(\mathbf{x})$ trained on data distribution $p_0(\mathbf{x})$ can sample by solving the reverse time diffusion SDE:

$$d\mathbf{x} = [f(t)\mathbf{x} - g(t)^2\mathbf{s}_\theta(\mathbf{x}, t)]dt + g(t)d\bar{\mathbf{w}} \tag{13}$$

where $f(t), g(t)$ are the time-dependent drift and diffusion coefficients, $d\bar{w}$ is the standard Wiener process. CNS solves the following SDE instead:

$$d\mathbf{x} = [f(t)\mathbf{x} - (\frac{g(t)}{\sqrt{k}})^2(k\mathbf{s}_\theta(\mathbf{x}, t))]dt + \frac{g(t)}{\sqrt{k}}d\bar{\mathbf{w}} \tag{14}$$

Practically, CNS scales the stochastic noise added at each sampling step by $1/\sqrt{k}$. When $k > 1$, less noise is added and the process generates samples with reduced variance, and vice versa. To analyze the relationship between CNS and temperature scaling, we denote the temperature-scaled data distribution $q_0(\mathbf{x})$, such that $q_0(\mathbf{x}) \propto p_0(\mathbf{x})^k$.

**Theorem C.1.** *For general data distribution $p_0(\mathbf{x})$, there is no prior distribution $q_T'(\mathbf{x})$, such that equation 14 starts from $q_T'(\mathbf{x})$ and generate the temperature scaled distribution $q_0(\mathbf{x}) \propto p_0(\mathbf{x})^k$.*

*Proof.* We start by considering the following forward SDE:

$$d\mathbf{x} = f(t)\mathbf{x}dt + \frac{g(t)}{\sqrt{k}}d\mathbf{w} \tag{15}$$

Let the initial distribution at $t = 0$ be $q_0(\mathbf{x})$, we define the time-dependent distribution generated by this forward SDE as $q_t(\mathbf{x})$. Then, one corresponding reverse SDE that can sample $q_0(\mathbf{x})$ takes the form of

$$d\mathbf{x} = [f(t)\mathbf{x} - (\frac{g(t)}{\sqrt{k}})^2(\nabla \log q_t(\mathbf{x}))]dt + \frac{g(t)}{\sqrt{k}}d\bar{\mathbf{w}} \tag{16}$$

Comparing equation 14 and equation 16, we can infer the following Lemma:

**Lemma C.2.** *The CNS reverse-time SDE equation 14 and the SDE equation 16 are equivalent if and only if $\nabla \log q_t(\mathbf{x}) = k\mathbf{s}_\theta(\mathbf{x}, t)$ for all time $t$.*

By construction, equation 16 evolves from $q_T(\mathbf{x})$ to $q_0(\mathbf{x})$. Now we assume CNS (equation 14) starts from the same prior distribution $q_T(\mathbf{x}) = \mathcal{N}(0, \frac{1}{k}\mathbf{I})$, by Lemma C.2, CNS correctly perform temperature scaling and sample from $q_0(\mathbf{x})$ if and only if $\nabla \log q_t(\mathbf{x}) = k\mathbf{s}_\theta(\mathbf{x}, t)$. Now we show that this condition is not true in general.

**Left Side**: To compute $q_t(\mathbf{x})$, we need to solve the SDE in equation 15. For an initial condition $x = X_0$, the solution $X(t)$ is given by the following stochastic interpolant:

$$X(t) = \alpha_q(t)X_0 + \sigma_q(t)\boldsymbol{\epsilon}, \quad \boldsymbol{\epsilon} \sim \mathcal{N}(0, \mathbf{I}) \tag{17}$$

$$\alpha_q(t) = \int_0^t f(s)ds = \alpha_t$$

$$\sigma_q(t) = \int_0^t \frac{g(s)^2}{k} \exp\left(-2\int_0^s f(u)du\right)ds = \frac{\sigma_t}{k}$$

Therefore, we can compute the $q_t(\mathbf{x})$ by

$$q_t(\mathbf{x}) = \int q_0(\mathbf{y})\mathcal{N}(\mathbf{x}; \alpha_t\mathbf{y}, \frac{\sigma_t^2}{k}\mathbf{I})d\mathbf{y} \tag{18}$$

**Right Side**. For the original diffusion process without scaling, we can compute the noisy distribution $p_t(\mathbf{x})$ at time $t$ as

$$p_t(\mathbf{x}) = \int p_0(\mathbf{y})\mathcal{N}(\mathbf{x}; \alpha_t\mathbf{y}, \sigma_t^2\mathbf{I})d\mathbf{y} \tag{19}$$

Comparing equation 18 and equation 19, we can infer that $\nabla \log q_t(\mathbf{x}) \neq k\mathbf{s}_\theta(\mathbf{x}, t)$ for general distribution. One simple counterexample is where $p_0(\mathbf{x})$ is a mixture of Gaussians. By previous reasoning, CNS cannot generate $q_0(\mathbf{x})$ if the prior distribution is $q_T(\mathbf{x})$.

What if we allow initial samples drawn from distributions other than $q_T(\mathbf{x})$? We consider the special case where $p_0(\mathbf{x}) = \mathcal{N}(0, \mathbf{I})$, then $p_t(\mathbf{x}) = p_0(\mathbf{x})$, $q_t(\mathbf{x}) = q_0(\mathbf{x})$. The condition $\nabla \log q_t(\mathbf{x}) = k\mathbf{s}_\theta(\mathbf{x}, t)$ trivially holds true. By Lemma C.2, CNS(equation 14) and equation 16 are equivalent. Therefore, CNS can generate $q_0(\mathbf{x})$ if and only if the prior distribution at time $T$ is the same as $q_T(\mathbf{x})$. For any other prior distribution, CNS would not be able to generate $q_0(\mathbf{x})$.

In conclusion, there does not exist an prior distribution $q_T'(\mathbf{x})$, from which CNS can always generate the temperature scaled distribution $q_0(\mathbf{x})$

$\square$

