# OpenReview forum: "Temporal Score Rescaling for Temperature Sampling in Diffusion and Flow Models"
_ICML.cc/2026/Conference — ICML 2026 regular_

### Official Review · Reviewer_AEDe · 2026-03-01

**Soundness:** 3
**Presentation:** 3
**Significance:** 3
**Originality:** 3
**Overall Recommendation:** 4
**Confidence:** 4

**Summary:**

This paper proposes TSR: a plug-and-play method to steer diffusion and flow models to sample from a locally sharper or flatter distribution than during training, without fine-tuning. The authors validate TSR on toy mixtures and then show consistent gains across several domains.

**Compliance With Llm Reviewing Policy:**

Affirmed.

**Final Justification:**

The rebuttal addressed most of my concerns, and despite remaining theoretical limits on real data, the method’s simplicity, breadth, and consistent low-cost gains justify a weak accept.

**Key Questions For Authors:**

* Can you give a clearer practical recipe for choosing (k, σ) (or σ alone) across schedulers, beyond grid search; ideally something based on η_t or a target rescaling profile?
* How sensitive are the reported gains to sampler choice and step count (especially for flow/ODE samplers where CNS is weak)? A compact robustness table would help.
* In at least one setting where likelihood is tractable (small model / low-dim), can you verify TSR actually matches the intended “tempered” target (even approximately), rather than just improving FID/CLIP?
* For the multi-task results, were baselines (esp. CNS) tuned to the same extent as TSR? If not, please add a fair tuning protocol or a sensitivity curve for each method.

**Limitations:**

You note that TSR only provides local scaling and theory is limited, but I’d encourage an explicit “failure modes” paragraph (e.g., when changing mode weights is necessary, when tuning is brittle, when TSR amplifies artifacts under strong guidance).

**Strengths And Weaknesses:**

## Strengths
1. Zero-training, essentially just scaling the score/velocity during sampling, and works with deterministic and stochastic samplers.
2. The SNR-based rescaling is clean in the Gaussian case, and the paper at least tries to bound the approximation error for well-separated mixtures.
3. It's rare to see a single inference tweak tested across image gen + protein + depth + pose + robotics, with generally sensible baselines.

## Weaknesses
1. "Temperature” is local, not global. TSR preserves mode weights (by design) and mainly changes intra-mode variance; this is fine, but the paper should be crisper about when local tempering is the right objective vs true global temperature scaling.
2. Guarantees are for simple distributions; on real data, TSR is heuristic. That’s okay, but some claims read stronger than what the theory supports.
3. The σ parameter is motivated intuitively, but in practice, users will tune (k, σ). I’d like clearer guidance or a cheap default-selection rule per sampler/model.
4. In some tasks, TSR uses “best (k, σ)”, while it’s not always clear whether CNS/other methods get equally careful tuning under the same compute budget.

---

> ### Author Rebuttal · Authors · 2026-03-31
>
> ## [Overview]
> We appreciate the reviewer’s helpful feedback, which guides us in improving the clarity of our paper. We thank the reviewer for highlighting the simplicity and broad applicability of our method. We address the reviewers concern below.
>
>
> ## [W1, W2]
>
> We thank the reviewer’s suggestion and will modify the main text accordingly.
>
> ## [W3+Q1] Hyper-parameter tuning
>
> $(k, \sigma)$ are agnostic to schedulers and usually only need to be tuned once for each model. As shown in ablations on multiple tasks (Figure 9, 12, and 14), sample performance follows a predictable, parabola-like curve as $k$ varies, with a fixed $\sigma$. Given such observations, we can fix $\sigma=1.0$ first, and find the corresponding optimal $k$ via binary search. We then tune $\sigma$ similarly while fixing $k$ to the found optimal value, and repeat the process once if necessary. This is much more efficient than a grid search.
>
> Additionally, this process doesn’t need to be fully optimal – a range of parameters gives similar performance, as shown in Figures 9, 12, and 14. Table 1 also confirms that $(k, \sigma)$ generalizes well across different models for the same task (e.g., SD3 vs SD2 vs Flux). Therefore, despite an additional $\sigma$ parameter, the tuning cost for TSR would be similar to the tuning cost of CNS and CFG.
>
>
> ## [W4+Q4] Tuning CNS
>
> Yes, for CNS, we perform a similar binary search over $k$ values as in TSR and report the optimal performance for each model. We will highlight this in the main text.
>
>
> ## [Q2] Robustness to sampling steps and samplers
>
> In Figure 10 in the appendix, we show TSR's advantage over CNS for various sampling steps ranging from 10 to 200. Table 5 in the appendix shows TSR’s advantage is consistent across DDPM, DDIM, and Euler-discrete samplers.
>
> ## [Q3] Verifying error in tractable setting
>
> We experiment with a tractable setting with a mixture of 6 random Gaussian modes with unequal weights. We measure the KL-divergence between the samples and the “tempered” target distribution (scaled either globally or locally). In both cases, TSR achieves a much smaller KL-divergence than CNS. The advantage is much more significant under the locally tempered target, aligning with our theoretical analysis.
>
> | Metric | CNS | TSR |
> |---|---|---|
> | KL-div v.s. globally tempered target | 0.4075 | 0.1688 |
> | KL-div v.s. locally tempered target | 0.3731 | 0.0501 |
>
>
> For more complex distributions but with tractable modes (e.g. the MNIST), we have also shown in Figure 17 that TSR maintains a good distribution of weights.
>
> ## [Limitation]
>
> We thank the reviewer’s suggestion and will add a paragraph on failure modes. Indeed, global scaling might be preferable when we want the most likely samples without caring about covering the modes. However, it remains an open question whether it is even possible to develop a global scaling method without extra training and NFE.

---

> > ### Author Rebuttal · Reviewer_AEDe · 2026-03-31
> >
> > Thank you for the detailed rebuttal. The rebuttal address most of my questions and make the paper stronger. My main remaining reservation is that the method is still more empirically justified than theoretically characterized on real data, and the $(k,\sigma)$ recipe is still somewhat heuristic. So while the rebuttal improves my confidence, I will keep my rating.

---

> > > ### Author Response · Authors · 2026-04-06
> > >
> > > Thank you for your thoughtful feedback and for taking the time to carefully consider our rebuttal. We are glad that our responses helped address most of your questions and strengthened your confidence in the work.
> > >
> > > We appreciate your point regarding the theoretical characterization on real data and the heuristic aspects of the method. Our goal in this work is to introduce a simple, practical mechanism that can be readily applied to existing models without additional training, and we agree that further theoretical understanding—especially in more realistic settings—is an important direction for future work. We will make sure to better emphasize both the current intuition and the limitations in the revised version.
> > >
> > > Thank you again for your constructive comments and careful evaluation.

---

### Official Review · Reviewer_wwJK · 2026-03-06

**Soundness:** 2
**Presentation:** 3
**Significance:** 2
**Originality:** 3
**Overall Recommendation:** 5
**Confidence:** 3

**Summary:**

This paper introduces a method that aims to sample from a temperature-scaled distribution of a flow-based generative model. The method is based upon the induced adjustment that is required to the drift in order to sample from a Gaussian with scaled noise. While the theoretical guarantees are limited beyond this setting, the resulting method incurs no additional inference cost as it amounts to a simple scaling of the drift. The authors validate the method in a range of applications, demonstrating that the it performs well despite its limited theoretical grounding, and compares favourably to the Constant Noise Scaling method (which is the main competitor given it also incurs no extra inference cost).

**Compliance With Llm Reviewing Policy:**

Affirmed.

**Final Justification:**

The authors' rebuttal and additional experiment resolved my concerns, so I have revised my score from 4 to 5.

**Key Questions For Authors:**

- Can the authors comment more on how $\sigma$ should be chosen in practice, and how the cost of choosing $\sigma$ would compare to the cost of choosing hyperparameters for CNS?

- Can the authors think of any experiments that could be done to compare performance against other methods for temperature scaling that incur additional computation, in order to more clearly quantify the benefits of their proposed zero-cost inference approach (or altenatively, can they justify why this would not be appropriate?)

**Limitations:**

Yes

**Strengths And Weaknesses:**

### Strengths
- The proposed method is simple, and can be used at no additional cost to standard inference. This sets it aside from most other methods which do have some additional overhead (though these other methods tend to have stronger theoretical grounding). Given that Constant Noise Scaling (CNS) (the main method to compare against) has been used in other works, having another zero-cost inference method that performs strongly relative to CNS is a good contribution.

- The authors assess the performance of the method in many different experimental settings, which I think are fairly convincing in allaying concerns regarding the limited theoretical grounding (I note that I am not too familiar with some of the experimental settings in the paper, such as the depth estimation and robotic manipulation, so I can't comment much on these experiments specifically).

- The authors include several ablations over the required hyperparameters $k$ and $\sigma$ in the Appendix.

- The paper is clear and well written, and is easy to follow and understand.



### Weaknesses
- The biggest weakness of the method is that it is only loosely grounded in theory. It is based on the behaviour of a simple Gaussian with rescaled noise, with some theoretical results controlling the error in the Gaussian mixture case; it is not necessarily clear how much this intuition will hold in more complex datasets. This limitation is inherent to the method, so unfortunately I'm not sure there's much the authors can do about this. In my opinion, they've done quite a good job of empirically validating that the method nevertheless behaves reasonably and is effective.

- The role of the hyperparameter $k$ is quite clear in my opinion, but the role of the $\sigma$ parameter is less so, given that it would ideally correspond to the variance of a targeted Gaussian but this is unknown in an applied setting. I thank the authors for their comments regarding this in section 4.2, while help explain $\sigma$'s effect on the weight schedule, and also for the ablations over $\sigma$ in the Appendix. However I think it is not necessarily clear how $\sigma$ should be chosen by a practitioner without having to run ablations over the possible values. Given that the proposed method relies on choosing both $k$ and $\sigma$, compared to choosing only a single parameter $k$ for CNS, I think it's important for practitioners to understand how easy it is to choose suitable values.

- The experimental comparisons are generally limited to only comparing to Constant Noise Scaling (and classifier-free guidance in the low-dimensional settings). I understand that this is because authors have only compared against other methods that incur no additional computation, but it makes it somewhat difficult to contextualise the results within the wider literature mentioned in the introduction. Given the paper's narrative is clearly focused on the zero-cost inference, I think the paper could maybe benefit from some comparison to an alternative method that does incur an extra cost, and to show how much additional cost would be incurred relative to any benefit gained.

---

> ### Author Rebuttal · Authors · 2026-03-31
>
> ## [Overview]
> We thank the reviewer for the thoughtful review and recognize the practical value of our work demonstrated via extensive experiments. To address the reviewer’s concern, we provide an efficient way for tuning $(k, \sigma)$ and additional comparison with the MCMC corrector baseline that utilizes additional inference-time computation.
>
> ## [W1] Theoretical bound
>
> As the reviewer points out, our extensive empirical results spanning multiple domains highlight that TSR is applicable in real-world settings. That said, even a theoretical bound based on mixture of gaussians is still meaningful as it still goes beyond what previous approaches (e.g. CNS) can provide.
>
>
> ## [W2 + Q1] $\sigma$ and hyper-parameter tuning
>
> As the reviewer points out, $\sigma$ serves as a control on how early the TSR scaling happens during sampling. The $(k, \sigma)$ pair **can be tuned efficiently**. As shown in ablations on multiple tasks (Figure 9, 12, and 14), sample performance follows a predictable, parabola-like curve as $k$ varies, with a fixed $\sigma$. Given such observations, we can fix $\sigma=1.0$ first, and find the corresponding optimal $k$ via binary search. We then tune $\sigma$ similarly while fixing $k$ to the found optimal value, and repeat the process once if necessary. This is much more efficient than a grid search.
>
>
> Additionally, this process doesn’t need to be fully optimal – a range of parameters gives similar performance, as shown in Figures 9, 12, and 14. Table 1 also confirms that $(k, \sigma)$ generalizes well across different models for the same task (e.g., SD3 vs SD2 vs Flux). Therefore, despite an additional $\sigma$ parameter, the **tuning cost for TSR would be similar to the tuning cost of CNS and CFG**.
>
>
> ## [W3 + Q2] Baseline with more compute
>
> We thank the reviewer for the suggestion. In the table below, we compare TSR against sampling with additional MCMC corrector steps, as proposed by Du et. al.. To quantify performance, we choose a tractable setting (mixture of 6 random gaussians) and measure the KL-divergence between the samples and the ‘ground-truth’ tempered distribution $p(x)^k$. It takes the MCMC method **~5 times** more score evaluations to surpass TSR.
>
> | Metric | CNS | TSR | CNS + MCMC (2x) | CNS + MCMC (5x) | CNS + MCMC (10x) |
> |---|---|---|---|---|---|
> | KL-div | 0.4075 | 0.1688 | 0.2937 | 0.1593 | 0.0837 |
>
>
>
> **Reference**
>
> Du, Y., Durkan, C., Strudel, R., Tenenbaum, J. B., Dieleman, S., Fergus, R., Sohl-Dickstein, J., Doucet, A., and Grathwohl, W. Reduce, Reuse, Recycle: Compositional Generation with Energy-Based Diffusion Models and MCMC. In ICML, 2023.

---

> > ### Author Rebuttal · Reviewer_wwJK · 2026-04-02
> >
> > I thank the authors for their response, which have done a good job of addressing my concerns. I like the new experiment comparing with additional MCMC correction steps, and I think including this in a revised version would certainly give more evidence that the method performs well at a low cost. Based on this, I have raised my score to 5 (I note again that I am not too familiar with some of the experimental settings though, so I keep my confidence at 3).

---

> > > ### Author Response · Authors · 2026-04-06
> > >
> > > Thank you for your thoughtful feedback and for taking the time to carefully consider our rebuttal. We are glad that the additional experiment with MCMC correction steps was helpful in addressing your concerns, and we appreciate your suggestion to include it in a revised version—we agree that it further supports the effectiveness of our method at low cost.
> > >
> > > We also appreciate your transparency regarding confidence,  and we are happy to further elaborate on any aspect if helpful.
> > >
> > > Thank you again for your support and constructive comments.

---

### Official Review · Reviewer_AH8s · 2026-03-11

**Soundness:** 4
**Presentation:** 4
**Significance:** 4
**Originality:** 4
**Overall Recommendation:** 5
**Confidence:** 3

**Summary:**

This paper introduces Temporal Score Rescaling (TSR), a training-free method designed to control the sampling diversity of denoising diffusion and flow matching models during the inference phase. The approach is based on the observation that applying a time-dependent rescaling to the score function of the noisy data distribution allows for "local" temperature control over the sampling distribution. Specifically, the authors analytically derive score rescaling formulas for isotropic Gaussian and mixture of Gaussian distributions. By applying a simple scaling factor $r_t(k,\sigma)$ to the score predictions of pre-trained models, TSR achieves efficient temperature sampling without requiring additional training or inference overhead.

The core contributions include: (1) proposing a universal, training-free sampling control framework applicable to any off-the-shelf diffusion or flow model; (2) demonstrating compatibility with both deterministic and stochastic samplers; and (3) validating the method's effectiveness across five distinct real-world tasks (image generation, depth estimation, pose prediction, robotic manipulation, and protein design). Empirical results demonstrate that TSR effectively reduces sample variance while maintaining mode coverage, leading to performance improvements across multiple benchmark datasets.

**Compliance With Llm Reviewing Policy:**

Affirmed.

**Final Justification:**

The authors' rebuttal effectively addresses my questions. This remains a solid paper with clear contributions, and I gladly maintain my positive score.

**Key Questions For Authors:**

1. In Section 4.2, $k$ is interpreted as a variance reduction factor, and $\sigma$ is presented as a proxy for the variance in the data distribution. However, in real-world applications, the true variance of the data distribution is unknown. Do you have recommended default parameter selection strategies, or a principled method for selecting the optimal $(k, \sigma)$ pair using a validation set?
2. The "local temperature scaling" discussed in the paper is distinguished from traditional global temperature scaling because it does not alter the weights of the components in a Gaussian mixture. Have you considered specific application scenarios where global temperature scaling might be more desirable, and would TSR still be applicable or adaptable in those cases?
3. In the protein generation experiments (Figure 6), TSR improves the designability score while maintaining diversity, but CNS appears to perform competitively in certain settings. How do you view the trade-offs between TSR and CNS across different application scenarios? Is there a principled way to determine which method to deploy based on the task?
4. Theorem C.1 proves that CNS cannot truly produce a temperature-scaled distribution. Does this theoretical result imply that CNS introduces systematic biases in certain practical situations? If so, how significant is this bias in practice?
5. In the Discussion (Section 6), you point out that TSR only alters "local" sampling and cannot change global mode weights. Have you considered extending the TSR framework to achieve true global temperature scaling, or is this a fundamental mathematical limitation of the current score rescaling approach?

**Limitations:**

**Yes.**

**Strengths And Weaknesses:**

* **Soundness:** The paper exhibits a high degree of technical rigor. The theoretical foundation is detailed, with precise derivations for score rescaling in simple scenarios (isotropic and mixture of Gaussians), complete with error upper-bound analyses. For Gaussian mixtures, the authors prove that the expected error is bounded across all timesteps and vanishes in extreme cases ($t \to 0$ or $t \to 1$). The experimental design is robust, spanning synthetic data to complex real-world tasks, utilizing appropriate evaluation metrics (e.g., FID, CLIP, AbsRel). However, the theoretical guarantees strictly apply only to simple distributions; a rigorous proof of convergence for complex, real-world data distributions is lacking, which constitutes a fair theoretical limitation.
* **Presentation:** The overall presentation quality is excellent. The paper is structurally clear, maintaining a logical flow from problem formulation and theoretical derivation to empirical validation. The mathematical derivations are exhaustive, and the figures are highly illustrative (Figures 1-3 provide an excellent intuitive understanding of the TSR mechanism). The literature review effectively contextualizes the work, clearly distinguishing TSR from existing methods like Classifier-Free Guidance (CFG) and Constant Noise Scaling (CNS). The supplementary materials provide extensive proofs and additional experiments, enhancing the paper's completeness.
* **Significance:** This research holds substantial practical importance. Temperature sampling is a highly sought-after capability in generative modeling. Existing training-free methods (like CNS) suffer from mode collapse and are incompatible with modern flow matching models and deterministic samplers. TSR offers an elegant, zero-inference-overhead solution. Furthermore, its robust validation across highly varied scientific and creative domains—particularly tasks requiring valid structural generation like automated protein design—strongly underscores its broad applicability and potential impact on applied machine learning.
* **Originality:** The proposed TSR framework is highly innovative. It introduces the first time-dependent score rescaling framework that frames temperature sampling as a time-varying scaling of the score function. Compared to pseudo-temperature sampling methods like CNS, TSR offers stricter theoretical guarantees and successfully maintains global distribution structures while controlling local variance. Additionally, TSR is orthogonal to CFG, allowing them to be combined for further performance gains. Translating this theoretical insight into a highly adaptable, plug-and-play technique is the work's standout original contribution.

---

> ### Author Rebuttal · Authors · 2026-03-31
>
> ## [Overview]
> We appreciate the reviewer's positive evaluation and recognition of the theoretical foundation, substantial practical importance, and originality of our work. We address the reviewer’s concerns and questions below.
>
>
> ## [W1] Theoretical guarantees
>
> We are glad the reviewer appreciates the theoretical foundation of our work, and agree that our theoretical guarantees are limited to mixture of gaussians. Nevertheless, prior approaches like CNS lack guarantees even under such settings. Our extensive empirical results spanning multiple domains highlight that TSR is applicable in real world settings.
>
> ## [Q1] Hyper-parameter tuning
>
> Yes, $(k, \sigma)$ can be tuned efficiently. As shown in ablations on multiple tasks (Figure 9, 12, and 14), sample performance follows a predictable, parabola-like curve as $k$ varies, with a fixed $\sigma$. Given such observations, we can fix $\sigma=1.0$ first, and find the corresponding optimal $k$ via binary search. We then tune $\sigma$ similarly while fixing $k$ to the found optimal value, and repeat the process once if necessary. This is much more efficient than a grid search.
>
> Additionally, this process doesn’t need to be fully optimal – a range of parameters gives similar performance, as shown in Figures 9, 12, and 14. Table 1 also confirms that $(k, \sigma)$ generalizes well across different models for the same task (e.g., SD3 vs SD2 vs Flux).
>
> Therefore, despite an additional $\sigma$ parameter, the tuning cost for TSR would be similar to the tuning cost of CNS and CFG.
>
>
>
> ## [Q2+Q5] Global v.s. local temperature scaling
>
> Global temperature scaling could be preferable when we want the most likely samples without caring about covering the modes (e.g. for depth estimation and robotics). However, it cannot be achieved by simply scaling the scores. (As a counter example, think of a mixture of two gaussians with unequal weights. A global scaling would change the weights, and thus change the direction of the score function.) Any training-free approach would likely require computing divergence of the distribution, and currently it remains an open question whether it is even possible to develop a global scaling method without extra training and NFE. As we show in experiments, TSR is applicable and effective for these tasks.
>
> ## [Q3] Choose between CNS and TSR
>
> In the protein experiments, TSR provides a superior trade-off between designability and diversity, producing points closer to the bottom-right region of the FID–designability plot (Fig.~6). More specifically,  for a given desired designability score, TSR generally achieve a lower FID score. For fairness, all TSR results use a fixed $\sigma$; further tuning of $\sigma$ can even further improve the trade-off for a given $k$.
>
> We agree that CNS can be competitive in some settings (e.g. pose estimation), but in general, TSR outperforms CNS (image, depths, protein, robotics) and has a broader applicability (e.g., deterministic sampling). Therefore, we believe TSR can be reliably chosen over CNS for most scenarios.
>
> ## [Q4] CNS bias
>
> Yes, CNS has a systematic bias of mode collapse, as shown in the checkerboard/swissroll distribution (Figure 3). As a simple practical setting, we quantify such bias in the MNIST experiment (Figure 17), where 40% of CNS samples fall under 1 class (out of 10 classes).

---

> > ### Author Rebuttal · Reviewer_AH8s · 2026-04-03
> >
> > Thank you for the rebuttal. The clarification on global vs. local temperature scaling was particularly helpful and addressed my question perfectly. I will maintain my original score.

---

> > > ### Author Response · Authors · 2026-04-06
> > >
> > > Thank you for your thoughtful follow-up and for taking the time to read our rebuttal. We are glad that the clarification on global vs. local temperature scaling was helpful and addressed your question.
> > >
> > > We appreciate your careful evaluation and consideration of our work.

---

### Official Review · Reviewer_By9z · 2026-03-15

**Soundness:** 2
**Presentation:** 3
**Significance:** 2
**Originality:** 3
**Overall Recommendation:** 4
**Confidence:** 4

**Summary:**

This paper proposes Temporal Score Rescaling (TSR), a training-free method to control the sampling diversity of pre-trained diffusion and flow matching models. Starting from the observation that for isotropic Gaussian data, the score of a variance-scaled distribution is a time-dependent linear rescaling of the original score, the authors derive a simple rescaling factor $r_t(k, \sigma)$ that can be applied to any score-based model's output at inference. TSR requires no retraining, adds zero inference cost, and is compatible with both deterministic (ODE) and stochastic (SDE) samplers. Experiments span five domains — image generation (SD3, SD2, FLUX), protein design (FoldingDiff), depth estimation (Marigold), pose prediction, and robot manipulation — demonstrating that TSR can steer sampling toward sharper or flatter distributions depending on the task.

**Compliance With Llm Reviewing Policy:**

Affirmed.

**Final Justification:**

The rebuttal provided by authors addressed my main concerns, and I raise my score to weak accept.

**Key Questions For Authors:**

1. Is it possible to derive a heuristic or taxonomy for choosing $k > 1$ vs. $k < 1$ based on task characteristics (e.g., generative vs. regression tasks, or some measurable property of the learned distribution)? Without such guidance, practitioners face a blind search over both the sign and magnitude of the temperature adjustment.

2. Has TSR been tested on step-distilled or few-step models such as consistency models, FLUX-turbo, or similar accelerated variants? These models are increasingly common in deployment and their score functions may not satisfy the MoG approximation that motivates TSR.

**Limitations:**

Yes

**Strengths And Weaknesses:**

### Strengths

- The method is remarkably simple — a single scalar multiplication on the model output at each timestep — yet well-motivated by a clean derivation from Gaussian assumptions. The theoretical progression from single Gaussian (exact) to MoG (bounded error, Theorem B.2-B.3) to general distributions (empirical) is clear.
- Broad experimental coverage across five disparate domains demonstrates the method's generality. The use of established pre-trained models (SD3, FLUX, Marigold, Pi-0) makes the results practically relevant.
- TSR is orthogonal to CFG and can be combined for further improvement beyond the CFG Pareto frontier (Figure 5), positioning it as a complementary technique rather than a replacement.
- The MNIST mode collapse analysis (Sec. A.5) provides direct quantitative evidence that TSR preserves mode coverage where CNS does not, supporting the "local temperature" claim.

### Weaknesses

W1: No principled guidance for hyperparameter selection, limiting practical usability.

The optimal $k$ direction differs across tasks (image generation uses $k < 1$; depth, pose, protein, and robotics use $k > 1$), with values ranging from 0.93 to 7.0. The paper does not provide any principle, heuristic, or taxonomy for how a practitioner should set $(k, \sigma)$ on a new task. For a method positioned as "plug-and-play," this means users still face a non-trivial hyperparameter search with no guidance on which direction to explore. This significantly limits the practical value of the contribution.

W2: Performance gains are modest and not consistently superior to CNS.

Across tasks, numerical improvements are small: FID improves by ~8% on SD3, AbsRel by ~6% on ETH3D, and robot success rate by +1.1%. On pose prediction, CNS (0.350) slightly outperforms TSR (0.356). The method's advantage over CNS is primarily in applicability scope (ODE samplers, flow matching models) rather than consistent performance superiority on shared ground.

W3: Generalization to distilled or few-step models is untested.

All experiments use full-step sampling (e.g., 30 steps for SD3/FLUX). Modern deployment increasingly relies on step-distilled models (e.g., consistency models, FLUX-turbo, SD3.5-turbo) whose score landscape differs substantially from full-step counterparts. Whether TSR remains effective in these settings, and whether the optimal $(k, \sigma)$ range transfers, is entirely unexplored.

---

> ### Author Rebuttal · Authors · 2026-03-31
>
> ## [Overview]
>
> We are glad the reviewer recognizes that our method is simple, theoretically well-motivated, widely applicable, and effective in avoiding mode collapse. We respond to the reviewer’s concern below. Specifically, we present a principled, efficient way for hyper-parameter tuning. We also explain how TSR can be incorporated to few-step model during distillation (similar to CFG).
>
> ## [W1 + Q1] Hyper-parameter tuning
>
> ### Determining the sign of k
> Yes, it is possible to determine whether to use $k>1$ or $k<1$ based on the task. For tasks requiring more accurate samples or samples with higher likelihood,  k>1 should be used. As we have shown in the paper, tasks such as depth estimation (more accurate estimation is preferred), protein folding (the protein would be desirable in the real world), robotic policy (task success benefit from the most likely action) fall into this scenario. For generative tasks (image/video generation), k<1 should be used as it encourages more details in the image/video, improving the image quality (aligned with the findings by *Karczewski et al.*)
>
>
> ### Tuning $(k, \sigma)$ efficiently
>
> **(a)** $(k, \sigma)$ can be tuned efficiently. As shown in ablations on multiple tasks (Figure 9, 12, and 14), sample performance follows a predictable, parabola-like curve as $k$ varies, with a fixed $\sigma$. Given such observations, we can fix $\sigma=1.0$ first, and find the corresponding optimal $k$ via binary search. We then tune $\sigma$ similarly while fixing $k$ to the found optimal value, and repeat the process once if necessary. This is much more efficient than a grid search.
>
> **(b)** Additionally, we don’t need to be fully optimal – a range of parameters gives similar performance, as shown in Figures 9, 12, and 14. Table 1 also confirms that $(k, \sigma)$ generalizes well across different models for the same task (e.g., SD3 vs SD2 vs Flux).
>
> **(c)** Finally, even for widely adopted inference-time methods like CFG, the optimal value varies across models, and a search is required.
>
> In summary, the tuning cost for TSR would be similar to the costs of CFG and CNS. We believe the requirement for tuning $(k, \sigma)$ does not diminish TSR’s value as a widely applicable plug-and-play method.
>
> ## [W2] Practical improvements
>
> As the reviewer points out, TSR generally improves performance and has a wider applicability (e.g., Deterministic sampling). TSR’s practical benefit also includes better mode coverage, as demonstrated by the protein folding and the MNIST experiment. While the performance advantage over CNS on some settings (pose estimation) are less substantial than others, TSR achieves them with zero training or additional inference cost. We believe a wide community of users can benefit from such performance gain with minimal effort (by using the efficient parameter-tuning scheme above).
>
>
> ## [W3 + Q2] Applicability to few-step models
>
> We thank the reviewer for pointing this out. Akin to CFG, TSR cannot be applied at test-time to an already distilled model (e.g. Flux-turbo, MeanFlow), but it can be applied at training/distillation stage, by simply distilling the distribution scaled by TSR. Since we focus on inference time temperature scaling, such experiments are left as future work. Additionally, there are domains where few-step model is not available or not strictly essential (e.g. protein design) and can benefit from our method.
>
> **Reference**
>
> Karczewski, R., Heinonen, M., and Garg, V. K. Devil is in the details: Density guidance for detail-aware generation with flow models. In ICML 2025

---

> > ### Author Rebuttal · Reviewer_By9z · 2026-04-04
> >
> > Thanks for the rebuttal of authors. My concerns about hyper-parameter tuning are addressed and it's good to know that the improvement is robust. I'll raise my score to weak accept.

---

> > > ### Author Response · Authors · 2026-04-06
> > >
> > > Thank you for your thoughtful feedback and for indicating that your concerns have been addressed. We really appreciate your time and consideration, and we are glad to hear that the clarification on hyper-parameter tuning was helpful.
> > >
> > > We noticed that you mentioned **raising your score to weak accept**, but the score may not have been updated in the system. We just wanted to gently check in case this was unintentional.
> > >
> > > Thank you again for your support and for the constructive discussion.

---

### Decision · Program_Chairs · 2026-04-30

**Decision:**

Accept (regular)

**Comment:**

This paper proposes a training-free method to steer the sampling diversity of denoising diffusion and flow matching models. By using a time-dependent linear rescaling of the original score, the proposed method can allow users to sample from a sharper or broader distribution than the training distribution. While some minor concerns remain at the end of the rebuttal period, overall reviewers and AC found the idea interesting, the demonstration mostly convincing, and the contribution worthy publishing in ICML.